# Carboxy-terminal polyglutamylation regulates signaling and phase separation of the Dishevelled protein

Marek Kravec [iD][1], Ondrej Šedo [iD][2], Jana Nedvědová [iD][3,4], Miroslav Micka [iD][1,2], Marie Šulcová[1], Nikodém Zezula [iD][1], Kristína Gömöryová [iD][1], David Potěšil[2], Ranjani Sri Ganji [iD][2], Sara Bologna [iD][2], Igor Červenka[1], Zbyněk Zdráhal [iD][2], Jakub Harnoš [iD][1], Konstantinos Tripsianes [iD][2], Carsten Janke [iD][5,6], Cyril Bařinka [iD][3] & Vítězslav Bryja [iD][1✉]

## Abstract

**Polyglutamylation is a reversible posttranslational modification that is catalyzed by enzymes of the tubulin tyrosine ligase-like (TTLL) family. Here, we found that TTLL11 generates a previously unknown type of polyglutamylation that is initiated by the addition of a glutamate residue to the free C-terminal carboxyl group of a substrate protein. TTLL11 efficiently polyglutamylates the Wnt signaling protein Dishevelled 3 (DVL3), thereby changing the interactome of DVL3. Polyglutamylation increases the capacity of DVL3 to get phosphorylated, to undergo phase separation, and to act in the noncanonical Wnt pathway. Both carboxy-terminal polyglutamylation and the resulting reduction in phase separation capacity of DVL3 can be reverted by the deglutamylating enzyme CCP6, demonstrating a causal relationship between TTLL11-mediated polyglutamylation and phase separation. Thus, C-terminal polyglutamylation represents a new type of post-translational modification, broadening the range of proteins that can be modified by polyglutamylation and providing the first evidence that polyglutamylation can modulate protein phase separation.**

**Keywords** Dishevelled 3; Polyglutamylation; TTLL11; Noncanonical Wnt Signaling; Protein Condensates
**Subject Categories** Development; Post-translational Modifications & Proteolysis; Signal Transduction

## Introduction

Polyglutamylation is a reversible posttranslational modification (PTM) initially discovered on tubulin (Edde et al, 1990) that is catalyzed by several members of the tubulin tyrosine ligase- like (TTLL) protein family (Janke et al, 2005). Humans possess 13 TTLL homologs of which 9 are active glutamylases. According to the current understanding, tubulin polyglutamylation is initiated by the formation of the branching point by the addition of glutamate onto a γ-carboxyl group of a glutamate residue within the substrate, followed by the addition of multiple glutamates thereby generating secondary glutamate chains of variable lengths. Individual TTLL polyglutamylases show catalytic preferences either for the initial branching or the elongation of glutamate chains (Mahalingan et al, 2020; van Dijk et al, 2007). Moreover, individual enzymes also show preferences for different substrates such as α- or β-tubulin (van Dijk et al, 2007), or other, non-tubulin proteins out of which only a few have so far been identified (Regnard et al, 2000; van Dijk et al, 2008; Ye et al, 2018). Polyglutamylation is a reversible modification; deglutamylation is catalyzed by enzymes from the cytosolic carboxypeptidase (CCP) family (Kimura et al, 2010; Rogowski et al, 2010; Tort et al, 2014).

In our proteomic datasets, we discovered Dishevelled (DVL), a highly conserved regulatory protein in the Wnt signaling pathway, as a candidate binding partner of several TTLLs. Wnt pathways represent evolutionary conserved signaling modules that are required for multiple processes during the development and maintenance of homeostasis. The best-studied is the Wnt/β-catenin pathway (also referred to as "canonical") where DVL is necessary for membrane signalosome complex formation (Bilic et al, 2007; Kan et al, 2020) which triggers the disruption of cytoplasmic β-catenin destruction complex and subsequent stabilization of β-catenin. β-catenin is translocated into the nucleus where it activates T-cell factor/lymphoid enhancer factor (TCF/LEF) dependent transcription (Clevers and Nusse, 2012). Another important branch is the the Wnt/planar cell polarity (Wnt/PCP) pathway, which is responsible for polarized cellular orientation and cytoskeletal rearrangements during morphogenetic cell movements (Boutros et al, 1998; Butler and Wallingford, 2017; Devenport, 2014; Humphries and Mlodzik, 2018; Klingensmith et al, 1994; Koca et al, 2022; Wallingford et al, 2000). DVL acts as a central mediator in the

[1]Department of Experimental Biology, Faculty of Science, Masaryk University, Brno, Czech Republic. [2]Central European Institute of Technology (CEITEC), Brno, Czech Republic. [3]Institute of Biotechnology of the Czech Academy of Sciences, BIOCEV, Vestec, Czech Republic. [4]Department of Biochemistry, Faculty of Science, Charles University, Prague, Czech Republic. [5]Institut Curie, Université PSL, CNRS UMR3348, Orsay, France. [6]Université Paris-Saclay, CNRS UMR3348, Orsay, France. ✉E-mail: bryja@sci.muni.cz

transmission of Wnt signals, facilitating the assembly and localization of core PCP components at the cell membrane (Klingensmith et al, 1994; Wallingford et al, 2000). By interacting with various downstream effectors, such as RhoA, Rac1, and JNK (c-Jun N-terminal kinase) (Boutros et al, 1998; Harrison et al, 2020), DVL orchestrates the cytoskeletal rearrangements including cilia and centrosome positioning critical for the proper functioning of polarized cells (Hashimoto et al, 2010; Park et al, 2008). Three human DVL paralogs—DVL1, DVL2, and DVL3 act as common components for most, if not all, Wnt signaling branches (Gao and Chen, 2010; Wallingford and Habas, 2005; Wynshaw-Boris, 2012). DVL was proposed to act as a branching point between individual downstream signaling pathways. How this function is achieved is unclear, but it is expected that numerous DVL-interacting partners and modulation of DVL function by complex PTMs, primarily phosphorylation (Hanakova et al, 2019), have a key role. In addition to the role of DVL in Wnt pathways, it was shown to be localized in the basal bodies of cilia, affecting their positioning (Park et al, 2008), in the centrosome interacting with core centrosomal proteins (Cervenka et al, 2016; Park et al, 2008) or in the spindle poles and kinetochores during the mitosis (Kikuchi et al, 2010).

DVL and its interacting partners can form cytoplasmic puncta that have all the features typical for biomolecular condensates (Fiedler et al, 2011; Gammons et al, 2016; Schubert et al, 2022; Schwarz-Romond et al, 2007a; Schwarz-Romond et al, 2007b). In recent studies, endogenous DVL puncta were observed asymmetrically localized to the vegetal pole of sea urchin or sea star embryos as a part of an axis-defying event (Henson et al, 2021; Swartz et al, 2021b). In mammalian cells, endogenously tagged DVL2 formed condensates associated with centrosomal structures that were affected by cell cycle progression or Wnt signaling activity (Schubert et al, 2022). The puncta formation, mediated by the phase separation, has been shown to be regulated by phosphorylation and ubiquitination (Bernatik et al, 2014; Vamadevan et al, 2022). However, the dynamics of DVL phase transitions is not well-understood and was so-far studied mostly for phosphorylation mediated by casein kinase 1 (CK1) δ and ε (Harnos et al, 2019; Qi et al, 2017).

Following our observation of interactions between DVL and TTLL enzymes, we aimed at determining whether DVL is a substrate of polyglutamylation, and whether this is of biological importance. Testing all TTLL glutamylases, we found that TTLL11 can efficiently polyglutamylate DVL3. Surprisingly, the modification takes place via the addition of a polyglutamate chain at the α-carboxyl of C-terminal methionine of DVL3, and not, as so-far described as a branched (secondary) amino acid chain to the γ-carboxy group of the internal glutamate (Redeker et al, 1991). DVL3 polyglutamylation changed the interactome of DVL3, in cells lowered its propensity to undergo phase separation and increased the activity in the Wnt/PCP pathway in *Xenopus* embryos. Both C-terminal polyglutamylation and its impact on phase separation were reverted by the enzymatic activity of the CCP6 deglutamylase, suggesting that DVL3 polyglutamylation is a physiologically relevant regulatory mechanism.

# Results

## DVL proteins are polyglutamylated by TTLL11

We have recently identified several members of the TTLL protein family in the pull-downs of DVL3 (datasets partially published in (de Groot et al, 2014), see Dataset EV1), suggesting that TTLLs and

DVL can interact. To assess whether any of the TTLLs can polyglutamylate DVL we overexpressed all 13 mouse TTLL paralogs tagged with EYFP together with Flag-tagged human DVL3 and performed western blotting (WB) analysis (Fig. 1A,B). We detected a change in the electrophoretic mobility of DVL3 when overexpressed with TTLL11, which indicated that TTLL11 might have modified DVL3. Moreover, analysis of polyglutamylation by an polyE antibody, which specifically recognizes polyglutamate peptides comprising at least 3 glutamates (Rogowski et al, 2010), revealed a prominent specific band in the presence of TTLL11 (in addition to signal corresponding to tubulin that served as a positive control) at the size of DVL3 (Fig. 1B). This signal was even more prominent when we used human TTLL11 (hTTLL11), indicating a higher efficiency of the human ortholog. We thus decided to use hTTLL11 in all subsequent experiments (unless specified otherwise). Importantly, co-expression of DVL3 with ligase-dead hTTLL11 (E531G) did not produce any polyglutamylation signal, revealing that the modification depends on the enzymatic activity of TTLL11 (Fig. 1C). The same experiments were performed with DVL2 and yielded identical results (Fig. EV1A,B). Next, we performed co-immunoprecipitation (co-IP) of DVL3 and TTLL11 (Fig. 1D) and could demonstrate that both proteins are not only present in one complex, but that DVL3 is also robustly polyglutamylated (Fig. 1D, WB: polyE). Similar results were obtained with other DVL paralogs DVL1 and DVL2 (Fig. EV1C). To show whether TTLL11 polyglutamylates DVL at endogenous levels, we performed IP of endogenous DVL3 and DVL2 in the presence and absence of exogenous TTLL11. As shown in Fig. 1E, both endogenous DVL3 and DVL2 were polyglutamylated (Fig. 1E). Together, this demonstrates that DVL proteins are substrates of TTLL11. In the further steps, we focused on DVL3 in order to describe the molecular details and significance of DVL3 polyglutamylation.

## TTLL11 polyglutamylates DVL3 at the C-terminal methionine

To map the polyglutamylation site(s) of DVL3, we first performed domain mapping using a series of truncated versions of DVL3. All three human DVL paralogs have the same domain organization with an N-terminal DIX (Dishevelled, Axin), a central PDZ (Post-synaptic Density Protein-95, Disc Large Tumor Suppressor, and Zonula Occludens-1), and the most C-terminal DEP (Dishevelled, Egl-10, and Pleckstrin) domain. These conserved domains are connected and C-terminally extended by intrinsically disordered regions (Gao and Chen, 2010). As shown in Fig. 2A (for raw data see Fig. EV2A), the DVL3 C-terminus was essential for the polyglutamylation by TTLL11 as detected by the polyE antibody (IP Flag, WB PolyE). Posttranslational polyglutamylation is generated on glutamate residues of the primary peptide chain on tubulin (Edde et al, 1990) and also on the non-tubulin substrates like nucleosome assembly protein (NAP) (Regnard et al, 2000). Replacement of glutamate by the highly similar amino acid aspartate within the polyglutamylation site of tubulin completely abolished tubulin modification in *Tetrahymena*, an effect that can be explained by the structure of TTLL (Mahalingan et al, 2020; Thazhath et al, 2002). Since we have observed polyglutamylation of all DVL paralogs (Fig. 1), we mutated all glutamates (E) conserved in the C-terminus (corresponding to aa 496–716 of DVL3) of

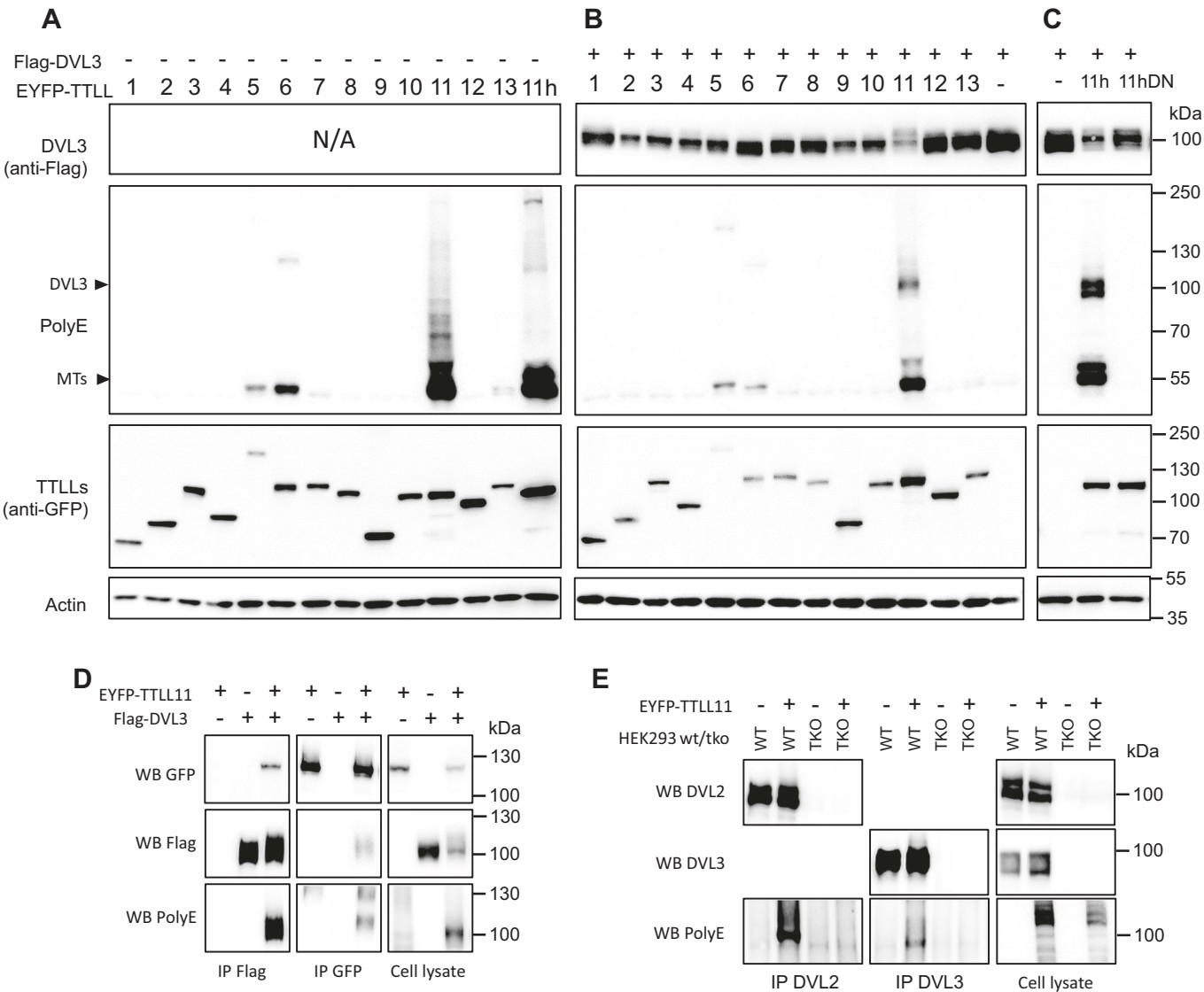

**Figure 1. TTLL11 binds and polyglutamylates DVL3.**

(A–C) HEK293T cells were transfected with constructs encoding murine EYFP-tagged TTLL1 – TTLL13 together with control plasmid (**A**) or DVL3 (**B**), or DVL3 with human TTLL11 and its inactive variant TTLL11 (E531G; DN) (**C**). The samples were subjected to WB analysis using polyglutamylation specific antibody—PolyE. Note the appearance of polyE-positive bands of DVL3 size in conditions where TTLL11 and DVL3 were co-expressed. The corresponding experiment with DVL2 is shown in Fig. EV1A,B. (**D**) Co-immunoprecipitation (co-IP) of Flag-DVL3 and TTLL11 overexpressed in HEK293T cells. TTLL11 is co-immunoprecipitated with DVL3; DVL3 in the pulldown was polyglutamylated when co-expressed with TTLL11 (IP Flag, WB PolyE). (**E**) IP of endogenous DVL3 and DVL2 from HEK293T cells in presence or absence of overexpressed TTLL11. Both endogenous DVL2 and DVL3 were polyglutamylated (WB PolyE). *DVL1/DVL2/DVL3* triple knockout (TKO) HEK293T cells served as a negative control. WB western blot, IP immunoprecipitation, MTs microtubules, WT wild type. Source data are available online for this figure.

DVL1, DVL2 and DVL3 into aspartates (D). Subsequently, we compared polyglutamylation levels of wild-type (WT) and E571D/ E604D/E693D/E710D DVL3 using the polyE antibody, and, strikingly, did not observe any differences (Fig. EV2B).

To identify the exact polyglutamylation site on DVL3, we used a mass spectrometry (MS) approach schematized in Fig. 2B. Analysis of tryptic digests of DVL3, which was co-expressed with TTLL11 in HEK293 and immunoprecipitated, by LC-MS/MS mapped polyglutamylation to the very last C-terminal DVL3 tryptic peptide— MAMGNPSEFFVDVM—with the most abundant peaks

corresponding to peptides with an additional 7–9 glutamate residues (Fig. 2Ca,b). Of note, the peptide contains three methionine residues (M703, M705, and M716) that get partially oxidized during sample processing. In Fig. 2C, we present signals for the fully oxidized peptide; for the experiment where we performed the analysis of all oxidation variants, see Fig. EV2C,D.

We hypothesized that, similarly to tubulin and NAP, glutamylation of this peptide is realized by branching and further polyglutamate chain extension at the sole internal glutamate (E710). Interestingly, however, the E710D DVL3 mutant was

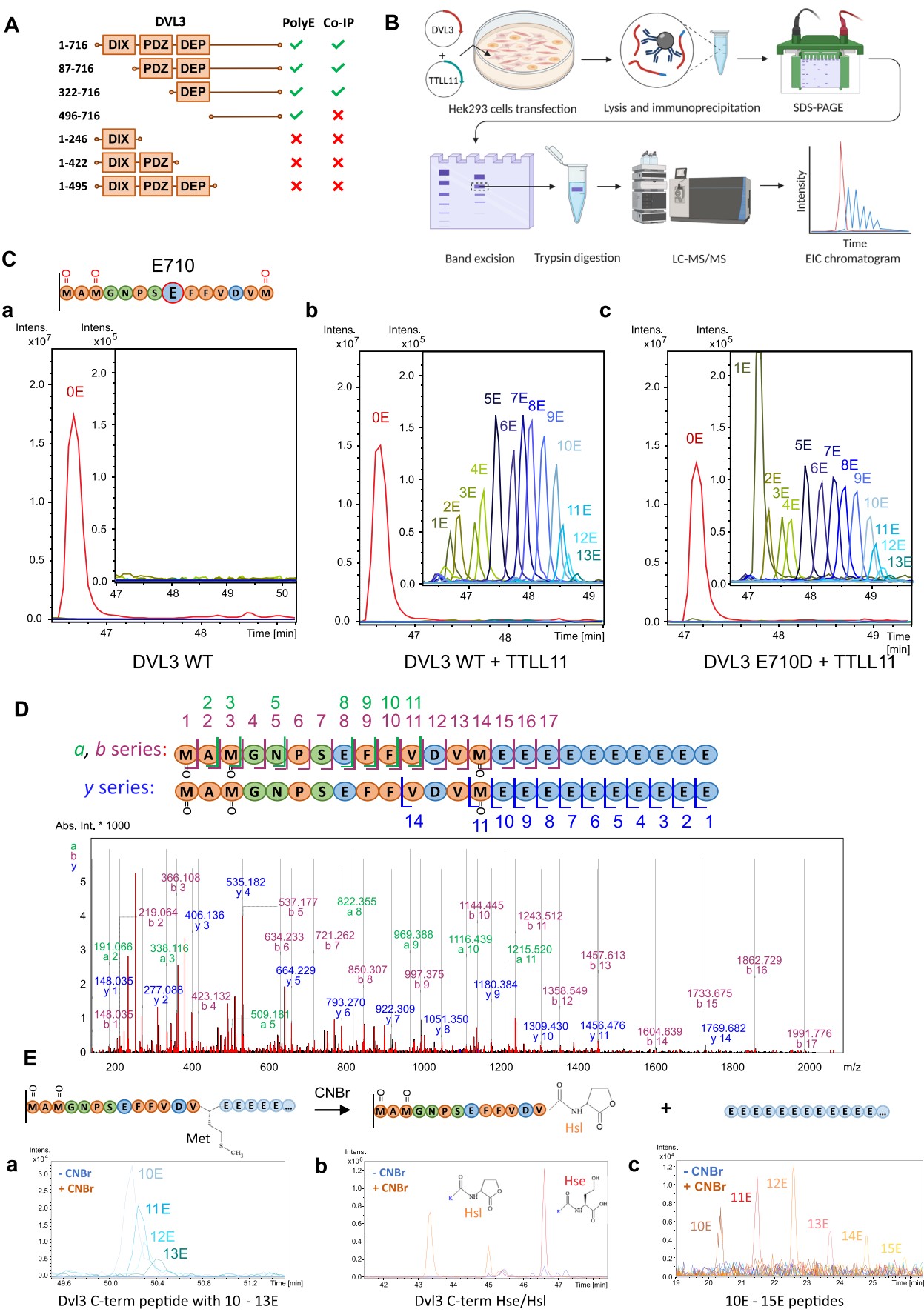

**Figure 2.  Modification site of DVL3 polyglutamylation and a novel character of the PTM.**

(**A**) Domain mapping of DVL3 polyglutamylation. DVL3 truncation mutants were co-expressed with TTLL11 in HEK293T cells, DVL3 was immunoprecipitated, and interaction with TTLL11 and DVL3 polyglutamylation was detected by anti-GFP and PolyE antibody, respectively. Raw data are shown in Fig. EV2A. (**B**) Scheme of the sample preparation for mass spectrometry (MS)-based analysis of polyglutamylation. (**C**) MS analysis of DVL3 polyglutamylation. The C-terminal tryptic peptide of DVL3 (schematized) was found to be polyglutamylated by TTLL11. EIC chromatogram shows peaks of this peptide in DVL3 (**a**), DVL3 co-expressed with TTLL11 (**b**) and DVL3 E710D in the presence of TTLL11 (**c**). Polyglutamylated variants (blue lines) which are zoomed in a separate window. Only peaks corresponding to 3x M-ox peptide are shown, all peaks for 0–3x M-ox C-terminal peptides are shown in Fig. EV2C,D. (**D**) MS/MS fragmentation analysis of polyglutamylated DVL3 C-terminal peptide (3x M-ox variant). The peaks in a, b and y series observed are highlighted by green, purple, and blue color respectively. MS/MS spectra of control synthetic peptides with 10E modification at M716 or E710 in comparison to the polyglutamylated DVL3 immunoprecipitated from cells is shown in Appendix Fig. S1. The aa sequences for individual fragments are shown in the schematic representation above the chromatogram. (**Ea–c**) Confirmation of C-terminal modification by CNBr cleavage. EIC shows peaks corresponding to polyglutamylated DVL3 C-terminal peptide before (blue) or after CNBr treatment (orange). C-terminal homoserine (Hse), homoserine lactone (Hsl), and free glutamate chains (10E–15E) are detected only after CNBr treatment. Hse or Hsl with polyE chain were not detected. The data shown correspond to 2x M-ox peptide; other M-ox variants are shown in Appendix Fig. S2A–D.

polyglutamylated to the same extent as DVL3 WT (Fig. 2Cc). Detailed analyses of peptide fragmentation by MS/MS suggested the presence of a linear chain of glutamic acid residues after C-terminal M716 of the peptide, which was visible, especially in *b*- and *y*-ion series (Fig. 2D). Such a modification, i.e., the elongation of the primary amino acid chain, was unexpected because it has never been reported before. Comparison of the MS/MS spectra of the C-terminal tryptic peptide of in-cell polyglutamylated DVL3 and synthetic peptides mimicking polyglutamylation either via branching at the internal E710 or by C-terminal elongation (Appendix Fig. S1) further supported this assumption. Among the MS/MS fragments of in-cell polyglutamylated DVL3, we did not find any fragments characteristic of branched peptide but observed several fragments unique for the linear polyglutamylated peptide. Altogether, our data strongly support the notion that TTLL11 catalyzes a so-far unknown type of polyglutamylation that extends the primary peptide sequence of DVL3 by the addition of multiple glutamates beyond the C-terminal methionine residue.

In order to directly prove C-terminal polyglutamylation at M716, we immunoprecipitated DVL3 that was co-expressed with TTLL11 and performed cyanogen bromide (CNBr) cleavage of the mixture of DVL3 tryptic peptides. CNBr cleaves peptide bonds at the C-terminus of non-oxidized methionine residues, releasing peptides with C-terminal homoserine (Hse) or homoserine lactone (Hsl) (Fig. 2E). After CNBr cleavage, the peak series corresponding to polyglutamylated MAMGNPSEFFVDVM disappeared (Fig. 2Ea), and instead non-glutamylated peptides terminated either by Hse or Hsl appeared (Fig. 2Eb). In addition, after CNBr treatment, we could detect signals corresponding to free polyglutamate chains of 10–15 glutamate residues (Fig. 2Ec). This effect was the most obvious in the double oxidized peptide (Fig. 2E), but the other oxidized forms of the C-terminal peptide showed similar results with the notable exception of the fully oxidized peptide that could not be digested by CNBr and thus conveniently served as a negative control (Appendix Fig. S2A–D). These results independently confirmed that DVL3 is polyglutamylated by the extension of the C-terminal M716 and provided further evidence for the existence of this novel type of PTM catalyzed by TTLL11.

## Sequence determinants of TTLL11-mediated DVL3 polyglutamylation

To determine molecular signatures of C-terminal polyglutamylation catalyzed by TTLL11, we generated a panel of DVL3 C-terminal mutants, where we (1) mutated the C-terminal glutamate residue 710 to aspartate (E710D); (2) mutated both acidic residues to alanines (E710A/D714A); (3) elongated the main polypeptide chain by the addition of four glutamate (4E) or four alanine (4A) residues; and (4) generated truncated variants lacking the C-terminal tail (DVL3 1–709; 1–697) (Fig. 3A). These mutants were then compared for their capacity to be modified by TTLL11. To this end, DVL3 was immunoprecipitated and polyglutamylation was analyzed by WB with the polyE antibody (Fig. 3B,C), as well as by MS-based detection (Fig. EV3A). Results from both experimental approaches were in good agreement (compare Figs. 3B and EV3A) and pinpointed several important facts. First, the modification is not strictly dependent on the presence of M716 or the exact C-terminal sequence as such because truncation mutants DVL3 1–697 and 1–709 as well as DVL3-4A were modified similarly to DVL3. Second, DVL3 E710A/D714A was not polyglutamylated at all and as such it could serve as a negative non-glutamylatable DVL3 control in subsequent functional experiments. Third, TTLL11 was clearly much more efficient in the polyglutamylation of DVL3-4E with the preexisting glutamate chain. To further study this effect, we generated DVL3 with only one extra glutamate (DVL3-E) at its C-terminus. As shown in Fig. EV3B already addition of one glutamate was sufficient to enhance polyglutamylation when compared to DVL3 WT.

In order to prove that TTLL11 directly polyglutamylates DVL3, we set up an in vitro polyglutamylation reaction using purified recombinant full-length DVL3 and TTLL11. We used DVL3 WT, DVL3-E, and the glutamylation-defective E710A/D714A mutant. Following in vitro incubation, the assay mixtures were analyzed by WB (Fig. 3D) and MS (Fig. 3Ea–d). Using WB, we could detect in vitro polyglutamylation by the polyE antibody only on DVL3-E. Interestingly, MS analysis not only confirmed polyglutamylation of DVL3-E but also detected monoglutamylated as well as polyglutamylated peptides in the DVL3-WT sample to a lower degree. This result suggests that TTLL11 itself is capable of DVL3 polyglutamylation but the ligation of the first glutamate can represent the rate-limiting step. It is possible that the in vitro assay with purified components is less efficient than polyglutamylation of DVL3 in cells, which could explain the difference in the number of added glutamates between these two assays. Notwithstanding this difference, our in vitro assay unambiguously demonstrates the direct and cofactor-independent modification of DVL3 by TTLL11.

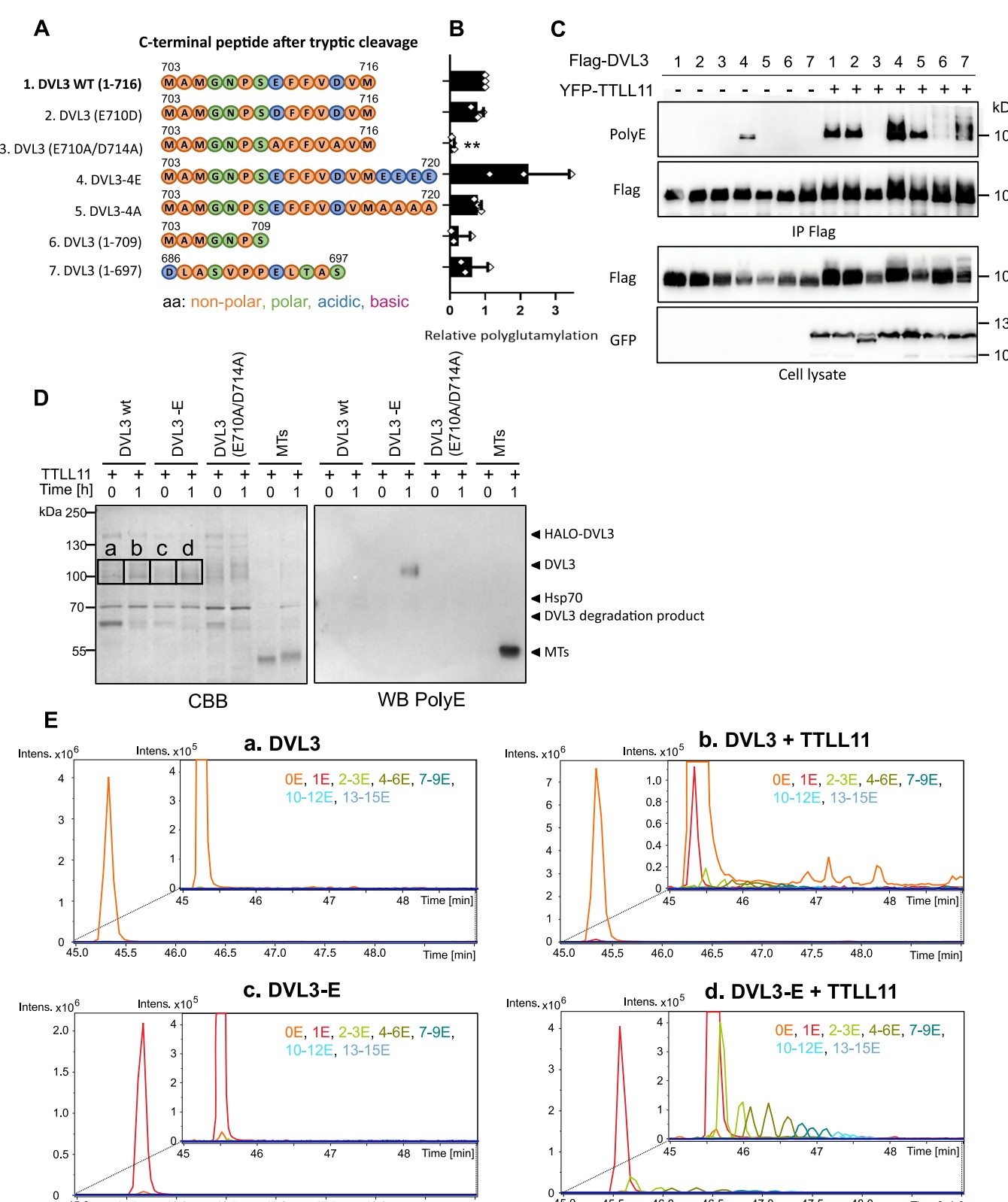

**Figure 3. Sequence determinants of TTLL11-mediated DVL3 polyglutamylation.**

(A) Schematic representation of DVL3 mutants. (B, C) HEK293 cells were transfected with the indicated Flag-DVL3 mutants and EYFP-TTLL11. DVL3 variants were immunoprecipitated and the extent of polyglutamylation was analyzed by anti-PolyE antibody ($n = 3$ biological replicates). Relative polyglutamylation is represented as PolyE band intensity, normalized to total protein (Flag) signal, and shown as a fold change compared to Dvl3 WT polyglutamylation. Statistical significance was assessed using one-sample $t$ test with theoretical median $= 1$; **$P = 0.0012$. Graph represents mean + SD. (D, E) In vitro polyglutamylation of purified DVL3/DVL3-E by TTLL11. The level of polyglutamylation was analyzed by SDS-PAGE followed either by western blotting analysis using polyE antibody (D) or bands corresponding to DVL3 (indicated as (a–d) on Coomassie brilliant blue (CBB) stained gel) were excised and used for subsequent MS analysis. (E) EIC showing detection of in vitro polyglutamylated DVL3 C-terminal peptide in the individual conditions (a–d). Source data are available online for this figure.

## Polyglutamylation changes DVL3 interactome

Polyglutamylation adds a significant negative charge to target proteins and it might regulate protein-protein interactions of DVL3 as proposed for polyglutamylation of microtubules (MT) (Janke and Magiera, 2020). We thus decided to analyze how the interactome of DVL3 changes upon polyglutamylation. We over-expressed WT or glutamylation-defective (E710A/D714A) DVL3 in the presence and absence of TTLL11, performed a pulldown of DVL3, and identified interacting partners by tandem MS (MS/MS) (for schematics, see Fig. 4A). The differences in the interactome of DVL3 and DVL3 polyglutamylated by TTLL11 are shown in the Volcano plot in Fig. 4B. Numerous proteins showed a change in the binding to polyglutamylated DVL3 in the presence of TTLL11. To restrict these putative hits to those that are most likely caused by DVL3 polyglutamylation and not by interaction with TTLL11, we performed an additional comparison using REPRINT with integrated CRAPOME and SAINTexpress tools (Fig. 4C). This bioinformatic pipeline allows comparison of multiple samples and efficiently removes false positives (Choi et al, 2012; Mellacheruvu et al, 2013). Proteins significantly changed in REPRINT analysis are highlighted in black in Fig. 4B. We have focused on those REPRINT patterns (Fig. 4C) where TTLL11 decreased/increased abundance in the pulldown of wild type but not glutamylation-defective E710A/D714A DVL3. Such polyglutamylation-dependent pattern identified several candidates whose interaction with DVL3 can be reduced (SRSF9, DAP3, GRWD1) or increased (KATNAL2, RAB11FIP5) upon DVL3 polyglutamylation. These proteins were not reported as DVL-interacting partners before (Sharma et al, 2018), which was, however, not the case for other hits—casein kinase 1 (CK1) δ and ε (CSNK1D and CSNK1E). These enzymes are known as major kinases regulating the functions of DVL (Bernatik et al, 2011).

## DVL3 polyglutamylation controls DVL3 activity in the Wnt/PCP but not Wnt/β-catenin signaling

We next aimed to determine the biological function of DVL polyglutamylation. DVL plays a key role in the activation of Wnt/β-catenin signaling, which induces T-cell factor/lymphoid enhancer factor (TCF/LEF)-dependent transcription. We thus first performed a series of experiments using TCF/LEF luciferase reporter (TopFlash assay) (schematized in Fig. EV4A). To test the impact of TTLL11 and DVL polyglutamylation on the Wnt/β-catenin signaling pathway. Interestingly, co-expression of TTLL11 with DVL1 (Fig. EV4Ba) or with DVL3 in combination with its activating kinase CK1ε (Kishida et al, 2001) (Fig. EV4Bb) efficiently reduced the TopFlash signal. Importantly, the luciferase signal

decreased upon co-expression of both, WT as well as catalytically inactive TTLL11 (E531G); (Fig. EV4B). This suggests that the excess of TTLL11 inhibits Wnt/β-catenin signaling independently of its enzymatic activity and represents an artifact. Indeed, TTLL11 (E531G) was as potent in inhibiting the TopFlash reporter as TTLL11 WT, even under conditions that bypass the need for DVL by expressing the constitutively active mutant of the key Wnt receptor LRP6 (LRP6ΔN) (Tamai et al, 2004) (Fig. EV4Bc,d). The mechanism of how TTLL11 inhibits Wnt/β-catenin is unclear, but this fact thwarted our efforts to determine the molecular function of DVL3 polyglutamylation by simply overexpressing TTLL11.

As an alternative approach, we studied the properties of DVL3 WT, polyglutamylation-mimicking DVL3 with 4 and 12 C-terminally added glutamates (DVL3-4E and DVL3-12E), and glutamylation-defective DVL3 (E710A/D714A). First, we have tested to what extent DVL3-4E/12E and DVL3 E710A/D714A can transduce signals induced by Wnt3a. Rescue assays in *RNF43/ZNRF3/DVL1/DVL2/DVL3 penta* knockout cells (DVL PKO) (Paclíková et al, 2021) suggested that, albeit slightly less efficient than DVL3 WT, both DVL3-4E and -12E can well mediate Wnt3a-induced signaling (Fig. EV4Ca). Non-modifiable DVL3 E710A/D714A mutant showed reduced, but still potent ability to transduce Wnt3a-induced signaling. However, this DVL3 variant was repeatedly observed at lower levels in comparison to DVL3 WT or DVL3-12E (see Fig. EV4Cb), and as such, we do not think that the result reflects the lower capacity of DVL3 E710A/D714A to transduce Wnt3a signal. In line with our observation that polyglutamylation does not affect the function of DVL3 in Wnt/β-catenin signaling, co-expression of DVL3-activating kinase CK1ε with both DVL3 WT and DVL3-12E potentiated TCF/LEF-dependent transcription to similar levels (Fig. EV4D). In addition, Wnt ligands were unable to modify DVL3 interaction with TTLL11 or TTLL11-induced polyglutamylation of DVL3 (Fig. EV4E). We thus conclude that polyglutamylated DVL3 is proficient in the Wnt/β-catenin signaling and DVL3 polyglutamylation does not directly affect DVL role in the canonical Wnt pathway.

To address the possible role of DVL3 polyglutamylation in the noncanonical Wnt/PCP signaling, we have used the well-established model of the *Xenopus laevis* embryos (Hikasa and Sokol, 2013). Wnt/PCP in the *Xenopus* model controls the process of convergent extension (CE) during gastrulation and neurulation. The role of candidate proteins in CE can be assessed by the injection of mRNA or inhibitory morpholinos (MO) (for experimental schematics and scoring system, see Fig. 5A,B. Injection of the *Xenopus* (x) Dvl3 mRNA affected CE but, strikingly, the polyglutamylation-mimicking variant xDvl3-12E triggered CE defects much more efficiently (Fig. 5C). To strengthen this observation, we have analyzed also an earlier developmental

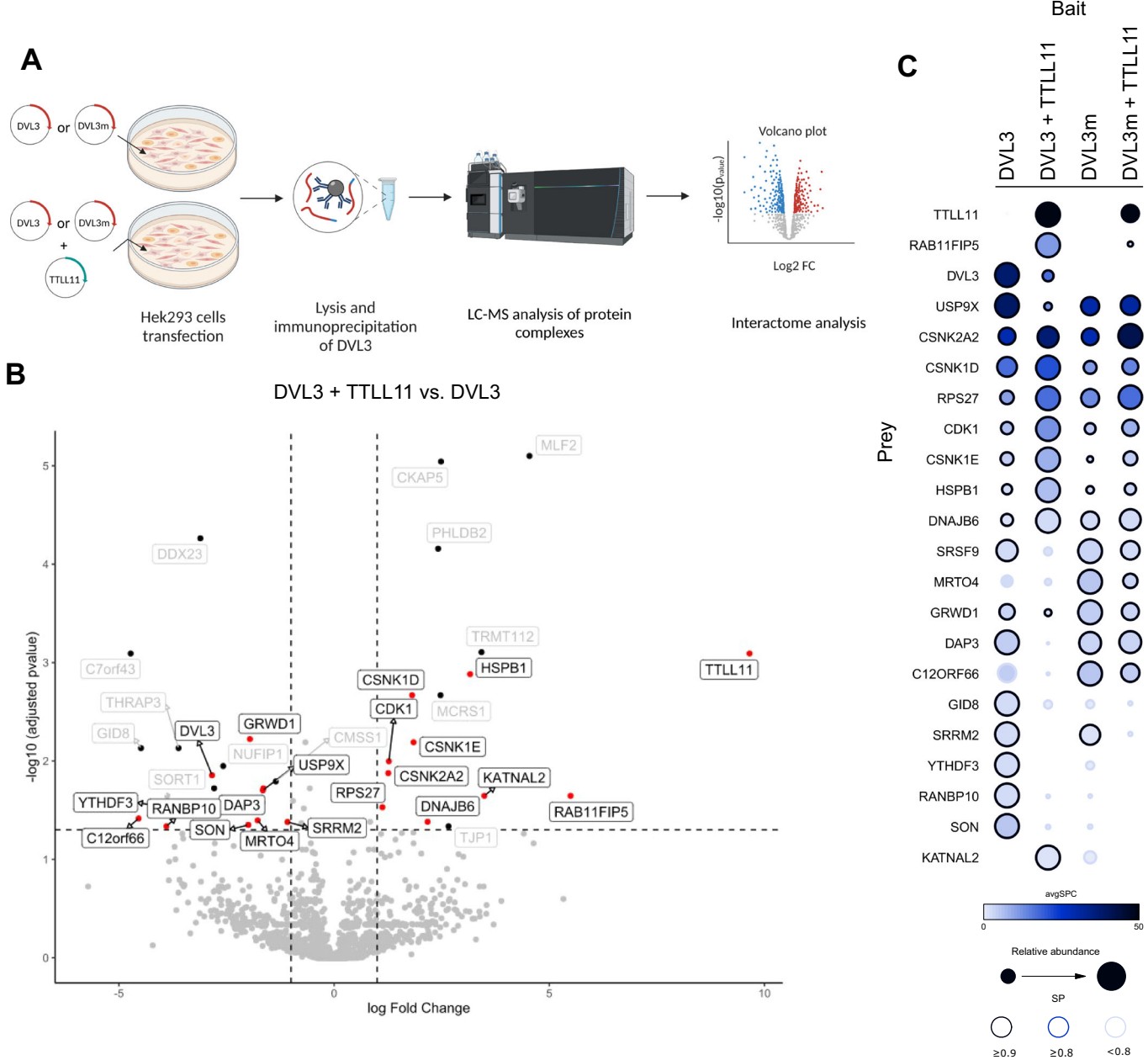

**Figure 4. Change of DVL3 interacting partners upon its polyglutamylation.**

(A) Scheme of sample preparation for MS analysis of DVL3 interactome upon co-expression with TTLL11. DVL3m = glytamylation defective DVL3 E710A/D714A. (B) Change of DVL3 interacting partners when overexpressed with TTLL11 was compared to overexpression of sole DVL3. DVL3 was immunopurified and interacting partners were analyzed by LC-MS in three biological replicates. The volcano plot shows detection of significantly upregulated and downregulated interacting partners using LIMMA statistical test—for more details, see "Methods" (FC ≥ 1, adjusted $P$ value < 0.05). (C) Hits from LIMMA were further analyzed by REPRINT with integrated CRAPOME and SAINTexpress tools; average spectra (AvgSpec); saint probability (SP). The significantly changed proteins are highlighted in black in the Volcano plot (B).

event—blastopore closure (Fig. EV5A). The blastopore closes at stage 11.5; but gets delayed when CE movements mediated by PCP signaling are disrupted (Fig. EV5B). In line with data in Fig. 5C, polyglutamylated xDvl3-12E was able to delay the blastopore closure to a larger extent than xDvl3 WT (Fig. EV5C). The WB analysis in the embryo lysates confirmed that the levels of individual xDvl3 variants were comparable (Fig. EV5D).

Interestingly, *xTTLL11* expression peaks exactly when CE movement during gastrulation starts (stages 8–11) in *X. laevis* embryos (Fig. EV5E,F). We have thus downregulated xTTLL11 by MOs targeting either its start codon (Fig. 5D) or splicing site (Fig. EV5G), which in both cases resulted in the defective CE.

The results in Fig. 5A–D suggest that TTLL11-mediated polyglutamylation of DVL is required for CE in *Xenopus*.

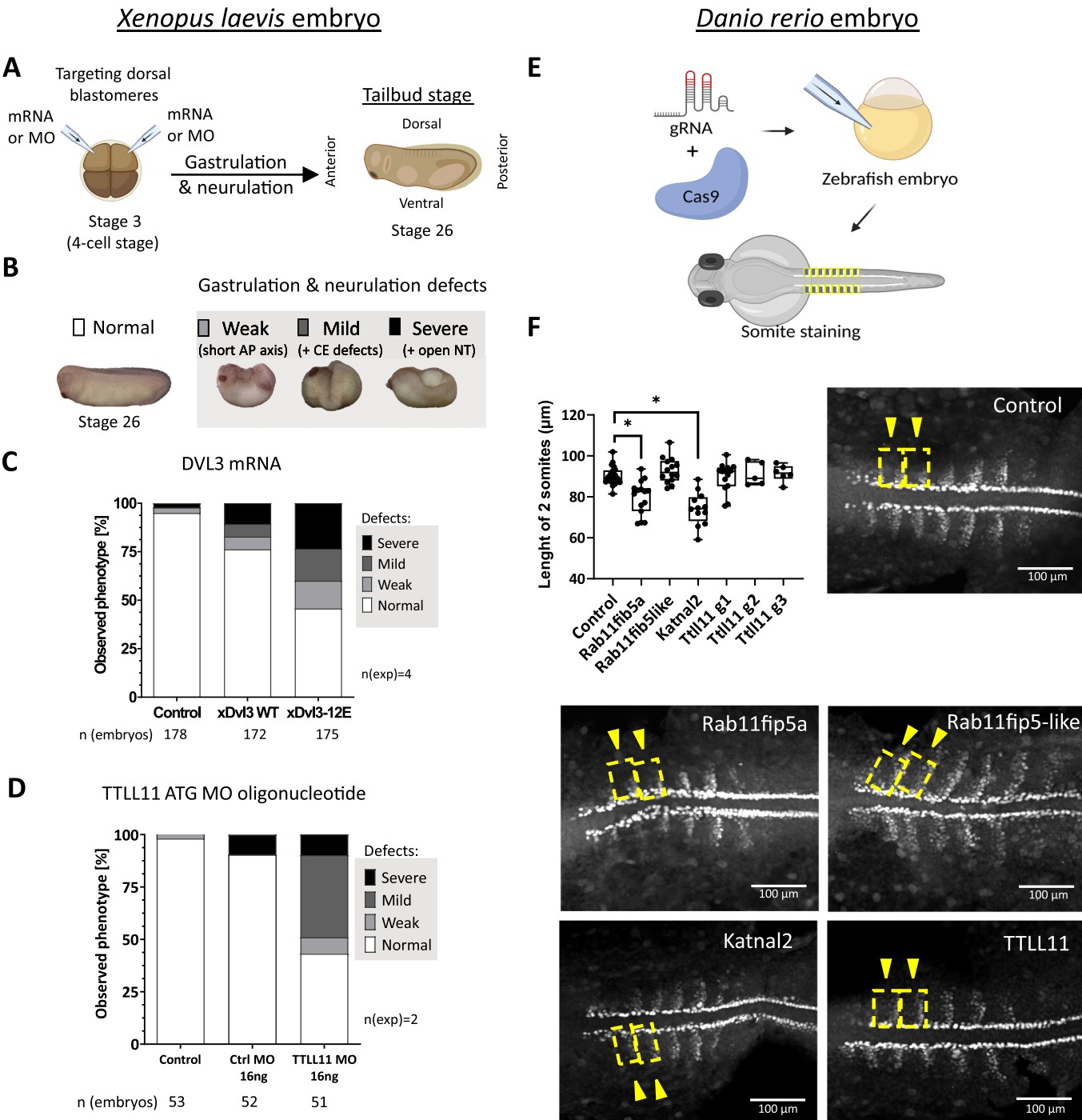

**Figure 5. Role of DVL3 polyglutamylation in *Xenopus laevis* and *Danio rerio* embryonal development.**

(A) Both dorsal blastomeres of 4-cell *Xenopus laevis* embryo were injected with xDvl3 mRNA or xTTLL11 morpholino (MO) and embryos were observed during gastrulation and neurulation at stage 26. (B) Phenotypes in *Xenopus* gastrulation and neurulation were assigned to four groups: no defects and weak (short AP axis), mild (+ convergent extension defects), and severe defects (+ open neural tube). (C) Effect of xDvl3 WT and xDvl3-12E modification mimicking mutant mRNA injection. (D) Effect of MO targeting xTTLL11 ATG start codon injection (for splicing MO see Fig. EV5G). (E) *Danio rerio* embryo was injected with Cas9 and gRNA targeting gene of interest. Embryos were fixed at 14.5 hpf and stained by α-MyoD1 antibody for somite visualization. (F) The length of the first 2 anterior somites was measured for control embryos and *Rab11fip5a, Rab11fip5-like, Katnal2,* and *TTLL11* KO embryos (n > 12). TTLL11 KO was additionally performed with two other gRNA TTLL11 g2 and g3 for verification of the result (n > 5). Box plots are shown as median (middle bar) with 25th and 75th percentiles and whiskers showing min to max values. Statistical analysis was performed by unpaired *t* test; *P < 0.05. Source data are available online for this figure.

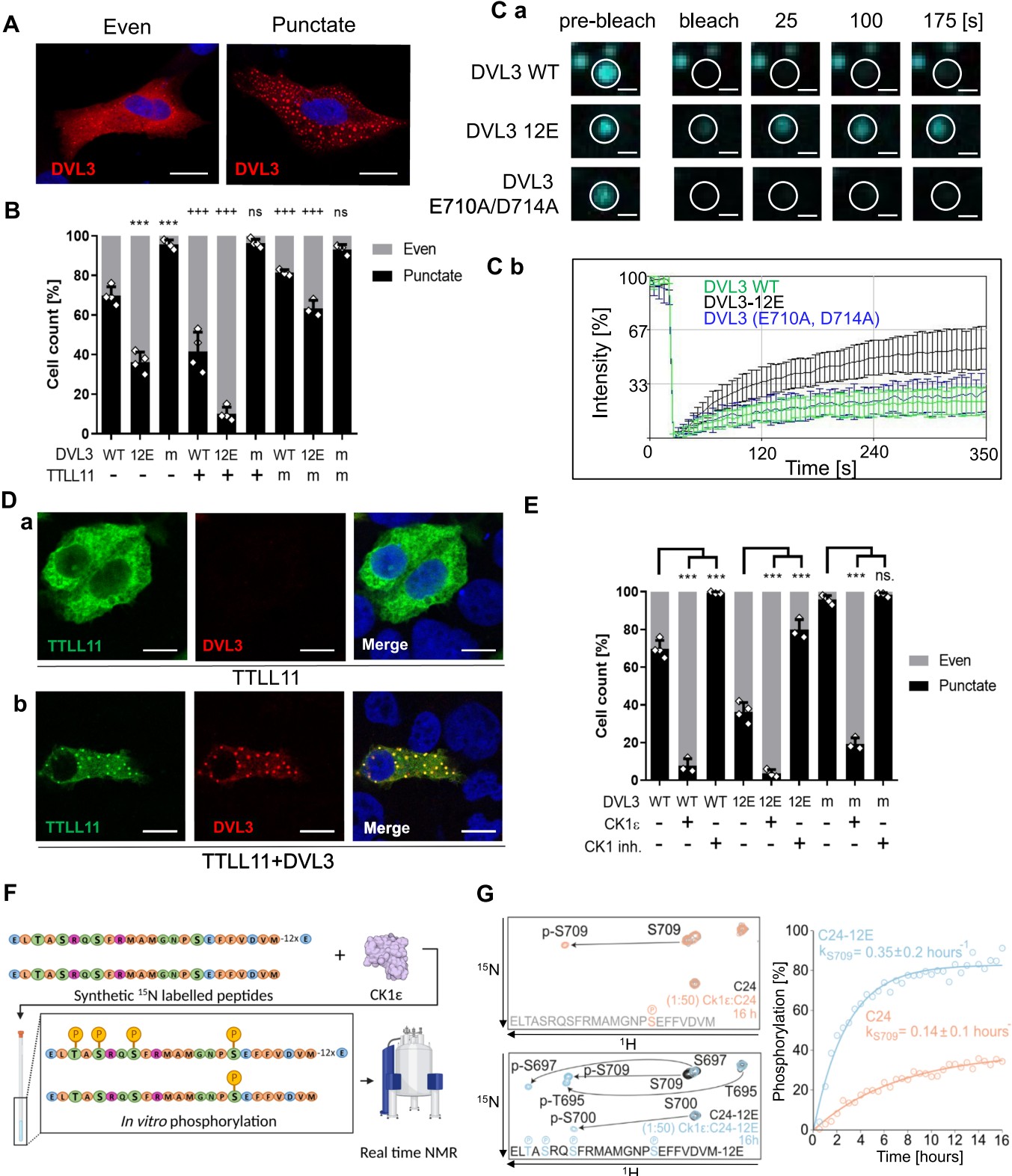

◄

**Figure 6. Polyglutamylation of DVL3 alters its phase separation and promotes phosphorylation by Ck1ε.**

(A) Overexpressed DVL3 in HEK293T cells. Scale bar = 10 μm. (B) Analysis of overexpressed DVL3 subcellular localization with its polyglutamylation variants in DVL1/DVL2/DVL3 triple KO HEK293 cells. The phenotype was assessed for >200 cells; n ≥ 3. Mean cell counts with SD are indicated. Statistically significant difference from WT DVL3 (*), or from the same DVL3 variant in the absence of TTLL11 or inactive TTLL11m (E531G) ( + ) is indicated. ***/+++ represent P < 0.001; ns not significant. DVL3m = DVL3 E710A/D714A (Ca) FRAP analysis of ECFP-DVL3 WT, ECFP-DVL3-12E and ECFP-DVL3 E710A/D714A intracellular condensates. (Cb) Intensities of fluorescence summarized in the graph. The data represent average from 15 condensates in three biological replicates (5 droplets/replicate and error bars represent SD). Scale bar = 2 μm. Raw data are shown in Appendix Fig. S3C. (D) Overexpressed TTLL11 in the absence (Da) or in the presence (Db) of co-expressed DVL3 in HEK293T cells. Scale bar = 10 μm. (E) Analysis of subcellular localization of DVL3 variants upon co-expression or inhibition of CK1ε by 1 μM PF670462. Phenotype was assessed for >200 cells; n ≥ 3 biological replicates. Mean values with SD are indicated. Statistically significant differences from the condition without CK1ε/CK1 inhibitor are indicated for each DVL3 variant (*). Experiments from 6B, 6E, and 7C were performed simultaneously, and control conditions are identical. For statistical analysis, see "Methods" and source data files. *** represent P < 0.001; ns not significant. (F) CK1ε in vitro phosphorylation of 24 DVL3-WT C-terminal aa (C24) or extended by 12E (C24-12E) peptides analyzed by NMR. (G, left) Overlay of HSQC spectra obtained before (gray and black for peptide C24 and C24-12E, respectively) and at the end of 16 h reaction (orange and blue for C24 and C24-12E, respectively). Arrows indicate chemical shift caused by phosphorylation. (G, right) Time course phosphorylation of C24 (orange) and C24-12E (blue) peptides at S709. The apparent rate constants were estimated from fitting to a single exponential function. Source data are available online for this figure.

Polyglutamylated DVL3 binds specifically KATNAL2 and RAB11-FIP5 (see Fig. 4). In order to test whether these downstream effectors also participate in CE, we have turned into zebrafish *Danio rerio* as a model. We have knocked out fish *Rab11fip5a*, *Rab11fip5-like*, and *Katnal2* genes as well as *Ttll11* by Crispr/Cas9 in zebrafish embryos (Fig. 5E) (Shah et al, 2015). Embryos upon deletion of *Rab11fip5a* and *Katnal2* showed body axis truncation phenotypes reminiscent of a defect in the PCP pathway (KO of *Wnt5b* served as the positive control, Fig. EV5H). This was further validated by quantitative analysis of the somite length at 14.5 hpf (Fig. 5F).

Altogether, these results suggested that polyglutamylated xDvl3 is more potent than xDvl3 WT in the activation of the Wnt/PCP pathway during CE movement. Consistently, downregulation of TTLL11 led to severe defects in gastrulation/neurulation on the *Xenopus* model, and KO of genes encoding interactors of the polyglutamylated DVL3 *Rab11fip5a* and *Katnal2* affected CE in *Danio rerio*. Collectively, this analysis supports the hypothesis that TTLL11-mediated polyglutamylation of DVL takes place during gastrulation to facilitate DVL-controlled morphogenetic processes such as CE movements.

## Polyglutamylation controls the phase separation of DVL3

What can be the biochemical features of DVL3 affected by polyglutamylation? Since DVL3 can form biomolecular condensates (Sear, 2007), we have addressed next this possibility. Depending on the extent of phase separation, DVL3 is distributed either homogeneously in the cytoplasm, or forms phase-separated "puncta" (Fig. 6A).

We thus overexpressed DVL3 WT, polyglutamylation-mimicking DVL3-12E, as well as the glutamylation-defective DVL3 E710A/D714A alone, or together with WT or ligase-dead TTLL11 (E531G) in HEK293 cells and analyzed the localization pattern of DVL3. To avoid interference with endogenous DVL proteins that were shown to affect this phenotype (Harnos et al, 2019) we have performed this experiment primarily in *DVL1/DVL2/DVL3* triple KO cells (Fig. 6B; the same experiment in WT HEK293 cells is provided for reference in Appendix Fig. S3A). Interestingly, polyglutamylation led to a decrease in DVL3 phase separation propensity, i.e., DVL3-12E was distributed more evenly in the cell cytoplasm compared to DVL3 WT (compare bars 1 and 2 in Fig. 6B). On the contrary, glutamylation-defective DVL3 E710A/

D714A localized predominantly into puncta (bar 3, Fig. 6B). Co-expression of TTLL11 further promoted even localization of DVL3 WT and DVL3-12E but not of glutamylation-defective DVL3 E710A/D714A (Fig. 6B, compare bars 4–6 with bars 1–3), which suggested that this phenotypic difference is indeed because of polyglutamylation. In line with this assumption, ligase-dead TTLL11 (E531G) promoted puncta formation and phase separation of DVL3 (Fig. 6B, bars 7–9). However, the downregulation of *TTLL11* by siRNA did not show any effect (Appendix Fig. S3B, C), pointing towards functional redundancy of TTLL11 with other endogenously present TTLL proteins. To address the effect of polyglutamylation on DVL3 phase separation more directly, we analyzed the dynamics of ECFP-tagged wild-type DVL3, DVL3-12E, and DVL3 E710A/D714A using fluorescence recovery after photobleaching (FRAP) (Fig. 6C, raw data in Appendix Fig. S3E). We have selected cells where DVL3 forms puncta, bleached individual puncta of comparable size (Fig. 6Ca), and quantified the signal for 325 s. As shown in (Fig. 6Cb) polyglutamylated DVL3-12E showed much faster recovery of fluorescence than DVL3 WT or DVL3 E710A/D714A. This suggests that the turnover of polyglutamylated DVL3-12E between phase-separated puncta and other cytoplasmic pools of DVL3 is increased. This observation agrees with its reduced phase separation propensity. Of note, overexpressed TTLL11 localized in cells rather uniformly to the cytoplasm and filamentous structures (probably MT) (Fig. 6Da) when expressed alone. When we co-expressed DVL3 and focused on cells that still show punctate localization of DVL3 (~35% of cells, see Fig. 6B) TTLL11 co-localized with DVL3 in puncta (Fig. 6Db). Similar behavior was observed for DVL1 and DVL2 puncta (Appendix Fig. S3D). This demonstrates that TTLL11 can be actively recruited to the sites where phase separation of DVL3 takes place and as such can regulate it in situ. Altogether, this data shows that polyglutamylated DVL3 has a lower tendency to phase-separate and consequently forms more dynamic biomolecular condensates.

## Polyglutamylation regulates phosphorylation of DVL3

The best-known regulators of DVL3 phase separation are several kinases—most notably CK1ε (Bryja et al, 2007) and centrosomal/ciliary kinases NEK2 and TTBK2 (Cervenka et al, 2016; Hanakova et al, 2019). These kinases upon binding and phosphorylation of DVL3 dissolve its condensates, resulting in more even cytosolic

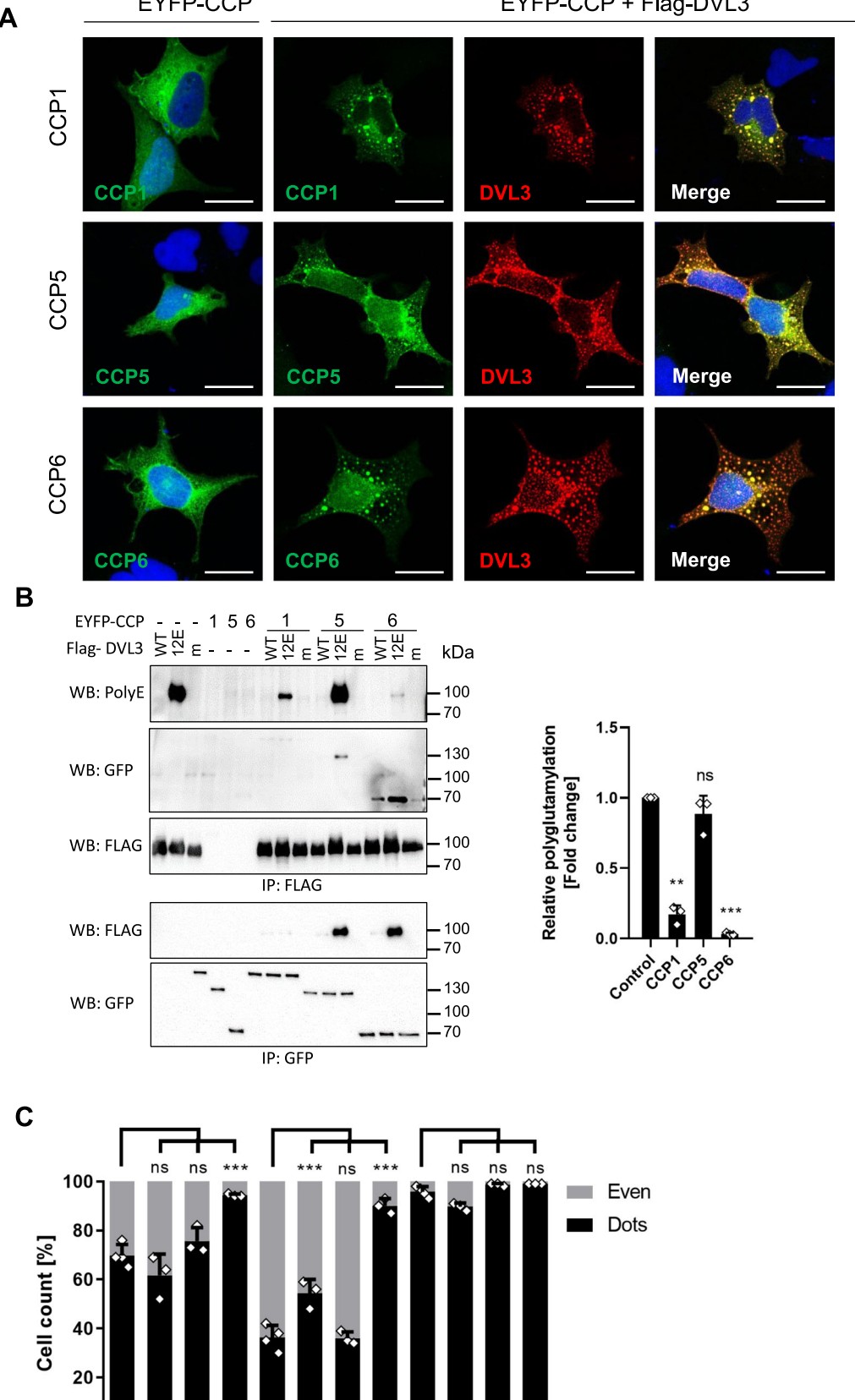

**Figure 7.   Deglutamylation of DVL3 by CCP enzymes.**

(A) Co-localization of overexpressed Flag-DVL3 with EYFP-tagged CCP1, CCP5 and CCP6 in RPE cells. CCPs are recruited from microtubule/even distribution into DVL3 puncta. DVL3 is visualized using primary anti-Flag and corresponding secondary Alexa Fluor 568 antibodies. Scale bar = 20 µm. (B) Deglutamylation and interaction of Flag-DVL3 by/with CCPs. DVL3 variants (DVL3m stands for E710A/D714A mutant) were co-expressed with EYFP-tagged CCP1, CCP5 and CCP6 in HEK293T cells, lysed, immunoprecipitated using anti-Flag antibody (DVL3) and anti-GFP (CCPs) and analyzed by western blotting (WB). Polyglutamylation was detected by modification-specific antibody PolyE, quantified using ImageJ software and normalized on immunoprecipitated DVL3 protein amount (Flag antibody) in three biological replicates. Statistics were calculated by a one-sample $t$ test with theoretical mean = 1. **$P$ = 0.002; ****$P$ < 0.0001, ns not significant (C) Cells were transfected as in (B) but in DVL1/DVL2/DVL3 TKO HEK293T cells and subcellular localization of DVL3 was analyzed by immunocytochemistry. Subcellular localization of DVL3 was assessed for at least 200 cells in ($n \geq 3$) biological replicates. Mean cell counts and SD are indicated. Statistical significance in comparison to the condition without CCP for each DVL3 variant is shown. **$P$ < 0.01; ***$P$ < 0.001, ns not significant. For statistical analysis, see "Methods" and source data files. Experiments from Figs. 6B,E and 7C were performed simultaneously, and control conditions are identical. Source data are available online for this figure.

distribution of DVL3 (Harnos et al, 2019). Interestingly, our interactome analysis (Fig. 4B,C) suggested increased interaction of CK1ε and CK1δ, a kinase closely related to and functionally redundant with CK1ε, with polyglutamylated DVL3. We further validated this observation by an independent approach—co-immunoprecipitation of endogenous CK1ε with overexpressed DVL3 in HEK293 cells was significantly promoted by co-expression with TTLL11 (Appendix Fig. S4A).

The obvious question then was "Are the polyglutamylation effects reflecting only the higher affinity of polyglutamylated DVL3 for CK1s or does polyglutamylation affect phase separation independently?". We have addressed the possible crosstalk between polyglutamylation and phosphorylation by the analysis of subcellular localization of DVL3 WT, polyglutamylated DVL3-12E, and glutamylation-resistant DVL3 E710A/D714A in the presence of exogenous CK1ε and in the presence of PF670462, the inhibitor of CK1δ and CK1ε. These experiments have been performed again in both *DVL1/DVL2/DVL3* TKO (Fig. 6E) and WT HEK293 cells (Appendix Fig. S4F). In both cases, co-expression of CK1ε strongly promoted uniform localization of all DVL3 variants, despite being less efficient in the case of DVL3 E710A/D714A (Fig. 6E). On the other side, CK1 inhibition, despite strongly promoting punctate distribution of DVL3, was not able to fully revert the even localization of DVL3-12E caused by polyglutamylation (Fig. 6E, bar 6). CK1 inhibition was less efficient when DVL3-12E was co-expressed with TTLL11 (see Appendix Fig. S4F). We conclude that both phosphorylation and polyglutamylation affect DVL3 phase separation independently, but their combination can be additive in vivo.

The observations of the additive effect of DVL3 C-terminal polyglutamylation and CK1ε-mediated phosphorylation raised the possibility that these events are mechanistically connected. To address this, we purified $^{15}$N-labeled DVL3 C-terminal peptides corresponding to aa 693–716 (C24) and aa 693–716-12E (C24-12E) and analyzed the CK1ε phosphorylation kinetics of individual sites in vitro by real-time NMR (Fig. 6F,G; Appendix Fig. S4B–E). DVL3 polyglutamylation modulates CK1ε-mediated phosphorylation pattern in a qualitative and quantitative manner. In common, both peptides get initially phosphorylated at S709, but only the polyglutamylated variant gets fully phosphorylated at this site due to the much faster kinetics under identical experimental conditions. The rest three sites resident at the C-terminal tail, T695, S697, and S700, are phosphorylated only in the polyglutamylated form of the peptide, although partially. The in vitro data provide evidence that polyglutamylation enhances CK1ε activity and significantly affects the phosphorylation dynamics of the DVL3 C-terminus. These two modifications can thus regulate each other and control DVL functions connected with the phosphorylation of the C-terminus—

including regulation of open and closed conformation (Harnos et al, 2019), participation in the noncanonical ROR2 signaling (Witte et al, 2010) or release from the centrosome during centriole separation (Cervenka et al, 2016).

## DVL3 polyglutamylation and polyglutamylation-induced changes in phase separation are reversed by CCP6

Polyglutamylation of tubulin can be reversed by deglutamylating enzymes from the CCP protein family (Kimura et al, 2010; Rogowski et al, 2010; Tort et al, 2014). We hypothesized that if polyglutamylation of DVL3 has a biological significance, then a deglutamylation mechanism should exist to control this function. CCPs, which have previously been shown to deglutamylate not only tubulin but also non-tubulin proteins such as myosin light-chain kinase (MLCK), whose C-terminus consists of polyglutamate stretch (Rogowski et al, 2010), were the most likely candidates for such function. We thus tested CCP1, CCP5, and CCP6 as the best-characterized deglutamylating enzymes. CCP1 and CCP6 are deglutamylases shortening polyE chains (Rogowski et al, 2010), while CCP5 was proposed to remove γ-linked branching glutamylation points (Rogowski et al, 2010), albeit it can also remove α-linked glutamates at a slower rate (Berezniuk et al, 2013). As shown in Fig. 7A, all three CCP enzymes were localized to the cytoplasm when overexpressed, however at least a fraction was recruited into DVL3 puncta upon DVL3 co-expression. In particular, CCP6 showed almost complete co-localization with DVL3 in these puncta. This suggests that CCP proteins might be capable of interacting with DVL3.

To evaluate such interactions and the potential deglutamylation activity on DVL3 directly, we co-expressed CCP1, CCP5 and CCP6 together with DVL3 WT, the polyglutamylation-mimicking DVL3-12E, as well as glutamylation-resistant DVL3 E710A/D714A (Fig. 7B). First, this experiment showed that CCP6 binds DVL3 WT more than CCP1 and CCP5 (IP: FLAG, WB: GFP). Second, we detected a strong complex between DVL3-12E and CCP5 and CCP6 but not CCP1 (IP: GFP, WB: FLAG). On the other hand, DVL3 E710A/D714A showed weaker interaction with all CCPs. This suggests that the interaction of CCP5 and CCP6 is promoted by the C-terminal polyglutamate sequence of DVL3. Lastly and most importantly, CCP1 and especially CCP6 were able to efficiently reduce polyglutamylation of DVL3-12E (IP: Flag, WB: PolyE), which agrees with their known function of removing C-terminal glutamate residues of a linear peptide chain. By contrast, CCP5, known to preferentially act on polyglutamylation branching points (Berezniuk et al, 2013; Rogowski et al, 2010), was less efficient in deglutamylating DVL3. The role of CCP1 and CCP6 in DVL3 deglutamylation was further confirmed by showing that

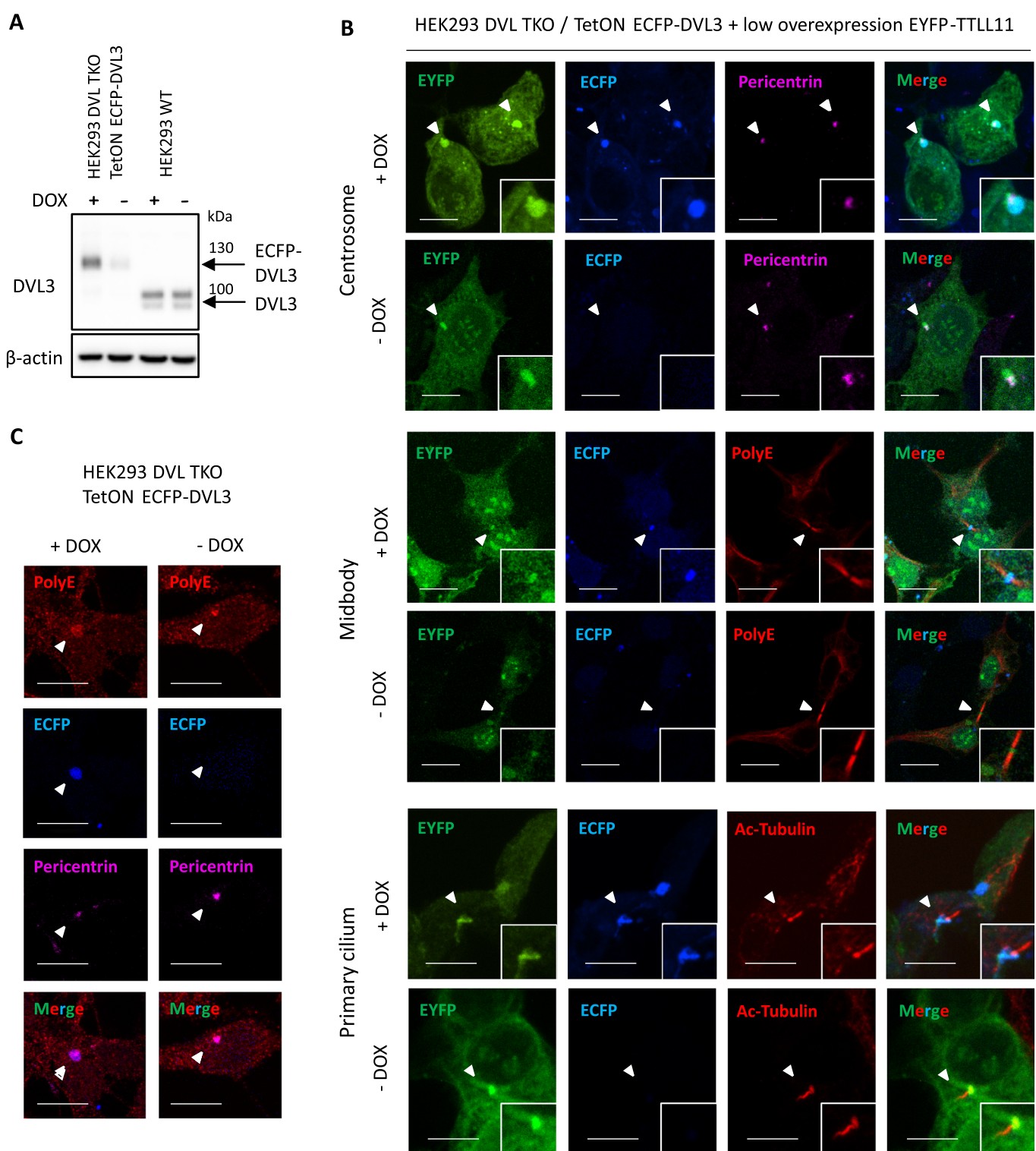

**Figure 8. Subcellular co-localization of DVL3 with TTLL11.**

(A) Stable cell line HEK293 DVL1/2/3 KO (triple KO–TKO) with inducible ECFP-DVL3 were transfected with low amount of EYFP-TTLL11 (20 ng) or control vector and doxycycline (DOX) induced near-endogenous level of ECFP-DVL3 was confirmed by WB analysis. (B) Co-localization of near-endogenous ECFP-DVL3 with TTLL11 in centrosome, midbody and primary cilium. Arrowheads point to the place of co-localization. (C) Polyglutamylation signal in in centrosome of cells with or without ECFP-DVL3. Arrowheads show centrosome. Micrographs shown are max-intensity projections of Z-stack taken to obtain whole microtubular compartments. Scale bar represents 10 μm. Source data are available online for this figure.

both enzymes, and in particular CCP6, can efficiently reduce DVL3 polyglutamylation generated by TTLL11 (Appendix Fig. S5A). This effect is specific and depends on the CCP enzymatic activity, as inactive mutants of both CCPs did not reduce DVL3 polyglutamylation. In summary, we showed that CCP1 and CCP6 can remove polyglutamylation of DVL3, and from the tested CCPs, CCP6 is the most efficient DVL3 deglutamylase. We propose that this specificity is achieved by the combination of its capacity to bind polyglutamylated DVL3 stronger than CCP1, which is also deglutamylating DVL3, and to remove linear polyglutamate chains (in contrast to CCP5, which also binds polyglutamylated DVL3).

Knowing that CCPs can remove polyglutamylation of DVL3 we asked whether they also revert the effects polyglutamylation has on phase separation of DVL3. We analyzed the subcellular localization of WT DVL3 and polyglutamylation-mimicking DVL3-12E after co-expression with CCP1, CCP5, and CCP6 (Fig. 7C; Appendix Fig. S5B). Interestingly, while CCP1 and CCP5 did not significantly change the localization pattern of DVL3, CCP6 was able to promote its punctate phenotype, suggesting it promotes the formation of DVL3 protein condensates. This effect of CCP6 was even more pronounced in the case of DVL3-12E (Fig. 7C, compare bars 5–8). This demonstrates that CCP6 not only deglutamylates DVL3 but also prevents its polyglutamylation-induced cellular phenotype. We conclude that CCP6 acts as an enzyme that removes DVL3 polyglutamylation and counteracts the role of polyglutamylation/ TTLLL11-mediated control of DVL3 phase separation.

### DVL3 co-localizes with TTLL11 in microtubular structures

The top hits that are binding specifically to polyglutamylated DVL3 (Fig. 4; RAB11FIP, KATNAL2) are proteins associated with microtubules and microtubule-formed structures. Therefore, we tested if we could detect co-localization of TTLL11 with DVL3 in structures such as centrosome, primary cilia and midbody (Fig. 8). We took advantage of previously characterized HEK293 *DVL1/2/3* triple KO with the inducible expression of ECFP-DVL3 (Cervenka et al, 2016). We optimized doxycycline induction to obtain the near-endogenous level of ECFP-DVL3 expression (Fig. 8A). Since we could find no antibody capably to specifically detect endogenous TTLL11, we co-transfected low amounts of EYFP-TTLL11 (Fig. 8B). Interestingly, we observed co-localization of both proteins in droplet-like structures that were associated with centrosome, basal body of primary cilium, or the middle part of midbody in case of dividing cells. We could not detect co-localization in mitotic spindle. The polyglutamylation detected by polyE antibody was too strong in case of EYFP-TTLL11 expression, but we could detect PolyE signal overlapping with DVL3 droplet surrounding the centrosome without the need to overexpress TTLL11 (Fig. 8C). We conclude that TTLL11 and DVL3 co-localize in several microtubule-based compartments and propose that this is the place where DVL3 polyglutamylation takes place.

## Discussion

### Novel type of PTM

Here we discovered that a key protein of the Wnt signaling pathway, DVL3, is post-translationally polyglutamylated by TTLL11 through a previously unknown and unique mechanism, in which a polyglutamate chain is generated at the α-COOH of the terminal amino acid of the substrate protein. The direct elongation of primary peptide chains by adding a variable number of glutamates substantially increases the breadth of polyglutamylation as a PTM. Further, many more proteins than previously expected can be polyglutamylated, because, as we demonstrated, it can take place at multiple C-terminal amino acids including M, A, E, and S. Moreover, TTLL11 could tolerate big changes at the DVL3 C-terminus—including deletion of C-terminal regions or C-terminal elongation by a stretch of 4 As. We propose that once TTLL11 interacts with its substrate it can behave as a rather promiscuous enzyme. Only one sequence modification completely prevented polyglutamylation in our hands: the mutation of two acidic residues, E710 and D714 to A, which made the DVL3 C-terminus very hydrophobic. It thus appears that the charge/hydrophilicity of amino acid residues near the C-terminal modification site, rather than the sequence motif or identity of C-terminal amino acid, is important for TTLL11-mediated polyglutamylation. This is in agreement with structural analyses of the catalytic site of TTLL enzymes (Natarajan et al, 2017) and their recognition domain (Garnham et al, 2015) which are both positively charged. Our discovery opens the exciting possibility that many other proteins get polyglutamylated by this mechanism. This is supported by the observation that overexpression of TTLL11 ((van Dijk et al, 2007), this study) resulted in multiple polyE-positive bands on the WB of whole-cell lysates. It remains to be identified if/what are the other substrates of TTLL11 in addition to DVL3, and to what extent can other TTLLs elongate α-carboxyl of other amino acids than glutamate and what are their substrate proteins.

### Novel biochemical/biophysical processes regulated by polyglutamylation

Given that the best-studied substrate of TTLL polyglutamylases is tubulin (Alexander et al, 1991; Edde et al, 1990; Redeker et al, 1991), initial insights into the function of polyglutamylation come from studies showing the role of this PTM in the modulation of interactions between microtubules and their interacting proteins, such as the microtubule severing proteins katanin and spastin (Lacroix et al, 2010; Shin et al, 2019; Valenstein and Roll-Mecak, 2016), or molecular motors (Sirajuddin et al, 2014). The identification of DVL3 as a polyglutamylation substrate allowed us to discover previously unknown, distinct functions of polyglutamylation, such as the control of protein phosphorylation and phase separation.

Our demonstration that polyglutamylation of DVL3 reduces its capacity to form biomolecular condensates phenocopies the effects of CK1ε—the established master regulator of DVL biology that controls its role in the Wnt signaling pathways (Bryja et al, 2007; Cong et al, 2004; Peters et al, 1999; Schwarz-Romond et al, 2005). These two mechanisms—polyglutamylation and phosphorylation of DVL3—probably act in parallel and are mutually supportive as CK1ε binds stronger to polyglutamylated DVL3 and polyglutamylated C-terminus of DVL3 gets phosphorylated by CK1ε much more efficiently. Both phosphorylation and polyglutamylation are adding a negative charge to the modification site. As charge distribution in IDPs significantly affects protein phase separation (Das and Pappu, 2013), disbalancing the amount and distribution of charged residues is likely the mechanism of polyglutamylation

and phosphorylation-mediated reduction in DVL3 phase separation capacity. Given the biochemical nature of polyglutamylation that allows for gradual charge and signal modulation—a regulatory principle so far only demonstrated for microtubule severing (Lacroix et al, 2010; Shin et al, 2019; Valenstein and Roll-Mecak, 2016)—we propose that polyglutamylation could play a unique role in fine-tuning of multiple phase separation processes where it can positively synergize with phosphorylation.

## Phase separation of DVL—physiological relevance and role in Wnt signaling

DVL clearly phase separates, but the physiological significance of this process is still unclear. In the overexpression setup, several key determinants that promote (DVL DIX domain, DVL DEP domain, ubiquitination by WWP2, but also several regions in the IDRs (Axelrod et al, 1998; Gammons et al, 2016; Kang et al, 2022; Schwarz-Romond et al, 2005; Vamadevan et al, 2022)) or reduce (phosphorylation by CK1 or NEK2, (Bernatik et al, 2014; Cervenka et al, 2016; Mund et al, 2015)) DVL phase separation has been identified. The fact that these determinants and stimuli do not directly correlate with the activity in the Wnt/β-catenin pathway suggests that the phase separation of DVL and its signaling activity in the Wnt/β-catenin pathway cannot be directly connected.

Till now several studies proved the existence of DVL protein condensates on the endogenous level. Many of them observed DVL condensates by immunostaining during the development of various organisms (Axelrod et al, 1998; Hawkins et al, 2005; Henson et al, 2021; Torres and Nelson, 2000; Yanagawa et al, 1995) and several studies observed that the distribution of DVL droplets was polarized (Miller et al, 1999; Swartz et al, 2021a; Torres and Nelson, 2000). Endogenous tagging of DVL and subsequent imaging of plasma membrane by TIRF microscopy suggested formation of DVL oligomers containing only several DVL molecules after pathway activation (Kan et al, 2020; Ma et al, 2020). Different group used super-resolution microscopy and observed small cytosolic condensates, that increased in amount after canonical Wnt stimulation (Kang et al, 2022). This suggests that a signaling complex at the plasma membrane, often termed as signalosome, which forms after Wnt stimulation involves DVL oligomerization, but is unclear if it involves phase separation. In noncanonical Wnt pathway, immuno-TEM and super-resolution imaging in sea urchin embryos observed that endogenous DVL forms dynamic droplets in cytosol, near the plasma membrane in the vegetal cortex as a part of cell polarization and embryo axis-defying event (Henson et al, 2021; Swartz et al, 2021a) Moreover, imaging in fly wing proposed dissolution of ectopic DVL condensates as part of PCP (Axelrod et al, 1998). Finally, immunocytochemical staining of endogenous DVL in model cell lines (Cervenka et al, 2016; Xie et al, 2018), C-terminal tagging of endogenous DVL2 (Schubert et al, 2022) as well as near-endogenous expression of DVL3-ECFP (Fig. 8) showed a cytoplasmic droplet that was associated with centrosome. Therefore, there could be different types endogenous DVL condensates, whose features need to be addressed in future. However, we propose that polyglutamylation affects the centrosome and microtubule compartment-associated DVL, where it co-localizes with TTLL11.

## Polyglutamylated DVL and microtubular compartments

DVL has been previously identified to be associated with several microtubular compartments such centrosome, basal body, cilium and midbody. One of the key functions of DVL is its role in the basal body and centrosome, summarized in (Bryja et al, 2017), where TTLL11 (van Dijk et al, 2007), as well as CCPs, including CCP6 (Rodríguez de la Vega Otazo et al, 2013) are also localized. Release of DVL3 from the centrosome is controlled by a centrosomal kinase NEK2 (Cervenka et al, 2016) that at the same time blocks phase separation of DVL3 (Hanakova et al, 2019). In the ciliated cells DVL, function in both, cell polarization and docking of basal body to the membrane (Park et al, 2008). DVL3 knockdown resulted in a significantly reduced ciliogenesis (Park et al, 2008; Zhang et al, 2023). Although we were unable to directly link DVL polyglutamylation and ciliogenesis it is intriguing that TTLL11 gets upregulated during primary cilium formation (Mathieu et al, 2021), RAB11FIP5 vesicles localize to the base of primary cilia (He et al, 2018) and KATNAL2 regulates ciliogenesis and brain development in *Xenopus* embryos (Ververis et al, 2016). Interestingly, RAB11FIP5 was shown to be necessary for the ciliary import of polyglutamylase TTLL5 (He et al, 2018). Given the fact that centrosome and pericentriolar matrix (PCM) is perceived as the phase-separated organelle (Zwicker et al, 2014), it will be exciting to explore to which extent polyglutamylation of non-tubulin proteins could, together with centrosomal kinases, control the dynamics of PCM clients via regulation of their phase separation propensity (Woodruff et al, 2017).

## Polyglutamylation in the noncanonical Wnt signaling

It has been shown earlier that decreased phase separation correlated with the "open" conformation of DVL3 (Harnos et al, 2019), which happened when binding of the DVL3 C-terminus to its own PDZ domain was prevented. Intriguingly, the binding of the DVL3 C-terminus to its PDZ domain is mediated by α-carboxyl of M716 (Harnos et al, 2019; Lee et al, 2015). Interestingly, it was shown in *Xenopus* embryos that the addition of the C-terminal tag to DVL, which is expected to lead to the open conformation, promoted the activity of DVL in Wnt/PCP but not Wnt/β-catenin (Qi et al, 2017). This is in perfect agreement with our observations and suggests that TTLL enzymes mediating physiological polyglutamylation can function as important regulators of CE during embryonal development. Peak expression of xTTLL11 at the onset of gastrulation as well as known functions of several proteins that specifically interact with polyglutamylated DVL3 support this possibility. In addition to CK1ε with the known role in the Wnt/PCP pathway (Bryja et al, 2008; McKay et al, 2001; Peters et al, 1999), we also showed that KATNAL2 and RAB11FIP5, affected CE in *Danio rerio* development. KATNAL2 has a role in microtubule severing, and its knockdown in *Xenopus* embryos led to defects in CE including blastopore and neural tube closure (Willsey et al, 2018). Moreover, the same study showed that KATNAL2 is expressed in neurogenic tissues and subcellularly localized to the basal body, ciliary axoneme, centriole, and mitotic spindle. RAB11FIP5, a RAB11 effector involved in apical endosome recycling pathway phenocopied Wnt11 in *Xenopus*, whereas its loss resulted in CE defects (Wallkamm et al, 2016). This indirect evidence suggests that TTLL-mediated polyglutamylation can integrate multiple

stimuli required for proper onset of activation of Wnt/PCP during gastrulation.

## Conclusions/limitations

In summary, we here discovered polyglutamylation via α-carboxyl of amino acid other than glutamate, which can effectively lead to the C-terminal linear elongation of primary peptide chains of a substrate. The minimal sequence requirements of the modification site make C-terminal polyglutamylation a potentially versatile PTM with a broad range of substrates. Analyzing C-terminal poly-glutamylation of the signaling protein DVL3, we discovered that polyglutamylation controls protein phosphorylation and formation of protein condensates. Polyglutamylation controls DVL3 activity in the Wnt/PCP pathway and early embryonal development. Our study keeps open several key questions that can become a basis for the follow-up work. First, TTLL11 is likely not the only enzyme that can catalyse DVL polyglutamylation. Downregulation of TTLL11 exhibited severe phenotype in *Xenopus*, while no phenotype was observed in *Danio rerio* (this study) or in mouse (see https://www.mousephenotype.org/data/genes/MGI:1921660; (Dickinson et al, 2016). The identification of other responsible TTLL enzymes can be challenging due to redundancy among individual TTLL family members that has been described earlier in other contexts, for example, for TTLL4 and TTLL6 (Casanova et al, 2015). Our study also does not provide a clear functional analysis in vivo using the rescue experiments. Generation of novel experimental models that will allow a detailed analysis of potential PCP defects in the best-defined tissue contexts such as polarity in the skin. Finally, because of the low levels of both DVL3 and TTLL11, the follow-up studies need to define the role of extent and function of polyglutamylated DVL at a completely endogenous level.

# Methods

## Cell culture, recombinant Wnt treatment, and near-endogenous expression of ECFP-DVL3

HEK293T (ATCC-CRL-11268), hTERT RPE-1 (kind gift from E. Nigg, Biozentrum, University of Basel, Basel, Switzerland), HEK293T-REx DVL1-2-3$^{-/-}$ and HEK293T-REx DVL1-2-3$^{-/-}$/RNF43$^{-/-}$/ZNRF3$^{-/-}$ (Paclíková et al, 2017; Paclíková et al, 2021) were grown at 37 °C and 5% CO$_2$ in complete DMEM medium containing 10% fetal bovine serum (DMEM; Gibco #41966-029), 2mM L-glutamine (Life Technologies, #25030024), 50 U/ml penicillin, and 50U/ml streptomycin (Hyclone-Biotech, #SV30010). Cells were routinely checked for mycoplasma contamination. Cell passaging of adherent cell lines was performed using trypsin (Biosera). The cells were washed in phosphate-buffered saline (PBS) buffer prior to the passage. For protein purification, suspension culture of FreeStyle™ 293-F Cells (Thermo Fisher Scientific) was cultured in FreeStyle™ 293 expression medium (Thermo Fisher Scientific) at 37 °C and 5% CO$_2$ while shaking at 120 rpm in Erlenmeyer cell culture flasks (Corning). For peptide purification, see "Peptide cloning, expression, and purification".

Treatment by recombinant Wnt3a and Wnt5a was performed as follows: HEK293T cells were treated with 1 μM Porcupine inhibitor LGK974 (Stem RD, #974-02) at the time of seeding and was maintained in the media whole time. Cells were transfected by corresponding vectors and 24 h post-transfection medium was supplemented with 100 ng/ml Wnt3a or Wnt5a (#1324-WN and #645-WN; R&D Systems) and cells were harvested after 3-h treatment.

For near-endogenous expression of ECFP-DVL3, stable cell line HEK293 DVL1/2/3 KO (triple KO–TKO) with inducible ECFP-DVL3 (Harnos et al, 2019) was treated with 0.5 μg/ml of doxycycline for 24 h. Primary cilia formation was achieved by changing cell media for a serum-free variant for 24 h. The samples for WB analysis of ECFP-DVL3 amount were prepared in the same way as the samples for immunostaining until the fixation step.

## Transient transfection

List of plasmids used in this study is available in Dataset EV2. Adherent cells were seeded with density of 30,000 cells per cm$^2$ if not indicated otherwise (in case of immunocytochemical methods the density was 10,000 cells per cm$^2$). They were incubated for 24 h and transfected using Polyetylenimin (PEI) with stock concentration of 1 mg/ml and pH 7.4. The amount of PEI and DNA as well as the incubation time of cells with transfection mixture differed according to experiment Dataset EV3. DNA and PEI were separately mixed with serum-free DMEM, incubated for 20 min at room temperature (RT) and subsequently mixed. Each mixing step was followed by vortexing and centrifugation. The final mix was incubated for 20 min at RT and added to the cells. A medium was replaced with the fresh complete DMEM after 4-h-long incubation with transfection mixture. The cells were harvested or fixed 24 h after transfection. Suspension FreeStyle™ 293-F cells were transfected in 200 ml culture with density of $4 \times 10^6$ cells per 1 ml. 400 μg DNA was diluted in 7 ml of PBS, while 1.2 ml of PEI (1 mg/ml) was diluted in 3 ml of PBS. Both mixtures were incubated for 20 min and subsequently mixed and incubated for another 20 min before adding to cell culture. In total, 200 ml of fresh media was added to the culture 4 h post-transfection.

## Dual-luciferase assay for analysis of Wnt/β-catenin signaling

Dual-Luciferase® Reporter Assay System (Promega) was performed by two reporter vectors. The test reporter vector (Super8X TopFlash) contained a gene for firefly luciferase controlled by the promoter containing TCF4-binding repeats. TCL/LEF together with β-catenin are co-activating subsequent transcription. The control reporter (pRLtkLuc) contained the gene for Renilla luciferase with the constitutively active promoter. Therefore, an activity of control reporter was used to normalize data for cell viability and transfection variability. The HEK293T cells were seeded on a 24-well plate and transfected according to Dataset EV3. For rescue experiment (Fig. EV4C), cells were treated with 1 μM Porcupine inhibitor LGK974 (Stem RD, #974-02) at the time of seeding. For Wnt stimulation, 200 μl of Wnt3a conditioned medium per condition was used in total 500 μl of medium per well for 14 h (as described in (Paclíková et al, 2021)). The cells were lysed after 24 h incubation (or 14 h in case of rescue experiment Fig. EV4C) and processed following the manufacturer's instructions. Luminescence was measured on MLX luminometer (Dynex Technologies).

## Site-directed mutagenesis and cloning

The Twin-Strep-Flag-HALO-DVL3 wt plasmid was made by Gateway Technology (via LR recombination according to the manufacturer's instructions; Thermo Fisher Scientific, #11791020) from the donor plasmid pDONR221 DVL3 (DNASU, #FLH178665.01x) and the destination plasmid pDEST-Twin-Strep-Flag-HALO (kind gift from Cyril Bařinka). The Twin-Strep-Flag-ECFP-DVL3 wt was made as follows: pdECFP-DVL3 (Harnos et al, 2019) vector was used as template for PCR product that was inserted into pDONR221, resulting in pDONR221-ECFP-DVL3 vector that was combined with pDEST-Twin-Strep-Flag-HALO via Gateway Technology (Thermo Fisher Scientific #11791020). HALO tag was subsequently removed by site-directed mutagenesis. Mutagenesis of pcDNA3.1-Flag-hDVl3, Twin-strep-Flag-HALO-DVL3 or pCS2+xDvl3 vector was performed using QuikChange II XL Site-Directed Mutagenesis Kit (Agilent Technologies) following a manufacturer's instructions. Some vectors were made by serial mutagenesis as shown in Dataset EV3. The PCR was performed using gradient thermal cycler (Bio-Gener Technology). For a bacterial transformation, One Shot® TOP10 Chemically Competent E. coli (Invitrogen) and Super Optimal Broth (SOC) medium were used (Thermo Fisher Scientific). The primers designed for mutagenesis reactions are listed in Dataset EV4. All mutations were verified using Sanger sequencing. For peptide cloning, see "Peptide cloning, expression, and purification".

## Quantitative PCR

RNA was purified by RNeasy Mini Kit (Qiagen). cDNA was synthesized following the manufacturer's instructions using the RevertAid Reverse Transcriptase kit (Thermo Fisher Scientific). qRT-PCR was conducted on a Roche LightCycler with the following program: initial denaturation at 95 °C for 10 min, followed by 45 cycles of 95 °C for 10 s, 60 °C for 30 s, and 72 °C for 1 s. Results were calculated as the fold difference between the target gene and the reference gene using the relative quantification $2^{-\Delta Cq}$ method. Transcript levels were measured using the LightCycler® FastStart DNA Master SYBR Green I kit (Roche) according to the manufacturer's protocol. 40S ribosomal protein S13 (RPS13) served as reference gene and primer sequences and probes are detailed in Dataset EV6.

## Co-immunoprecipitation

The HEK293T cells were grown at 10-cm dishes to 60–70% confluency. They were transfected with desired vectors and incubated for 24 h post-transfection. The following work was performed at 4 °C. Cells were lysed for 20 min in 1 ml of lysis buffer (50 mM Tris, pH = 7.4, 150 mM NaCl, 1 mM EDTA, 0.5% NP40) supplemented by 1 mM DTT (Sigma-Aldrich) and 1× protease inhibitors (Roche Applied Science). Crude cell lysate was cleared by centrifugation for 15 min at $16,000 \times g$. The 80 μl of total cell lysate (TCL) was mixed with 20 μl of 5× Laemmli buffer. The remaining supernatant was divided into samples and each sample was incubated overnight with 1 μl of the corresponding antibody. G-protein coupled sepharose beads (GE Healthcare) were pre-washed in lysis buffer and blocked overnight in the lysis buffer with 1% BSA. The next day, lysates were combined with 15 μl of the pre-washed G-protein coupled sepharose beads. The mixture was incubated on carousel for 4 h. After the incubation, the beads were washed 5× in lysis buffer and proteins were eluted by the addition of Laemmli buffer to total volume of 80 μl. Used antibodies are listed in Dataset EV5.

## Western blotting

For WB, the samples were incubated at 95 °C for 5 min and subjected to SDS-PAGE, then electrotransferred onto Hybond-P membrane (GE Healthcare), immunodetected using appropriate primary and secondary antibodies (conjugated with HRP) and visualized by ECL (Millipore) and documented using FusionSL system (Vilber-Lourmat). WB signal intensities were quantified using ImageJ. The area of the peak intensity for the protein of interest was divided by corresponding values of peaks intensity obtained for control protein. Used antibodies and corresponding dilutions are listed in the Dataset EV5.

## Immunofluorescence

The HEK293T cells were seeded onto 13-mm coverslips, coated in 0.1% gelatin. Next day, the cells were transfected with vectors according to the design of experiments and if applicable treated by 1 μM PF670462 (DC chemicals, #DC2086), or the medium was changed for serum-free variant, and subsequently incubated for 24 h. Then, cells were washed in PBS, fixed in 4% paraformaldehyde (PFA, Millipore) in PBS for 1 h followed by three washes in PBS and finally blocked in PBTA (PBS, 5% donkey serum, 0.3% Triton X-100, 1% BSA). Samples were incubated overnight at 4 °C with primary antibodies diluted in PBTA. Following three washes in PBS, samples were incubated with corresponding Alexa Fluor secondary antibodies (Invitrogen, Abcam) for 1 h at RT, followed by 5 min incubation at RT with DAPI (Thermo Fisher) for nucleolar staining. Finally, samples were mounted in DAKO KO mounting solution (DAKO KO). The images were taken using a fluorescent (Olympus IX51) or confocal (Leica SP8) microscope using ×40 water or ×60 oil objectives. For protein subcellular localization experiments, at least 200 positive cells per experiment (n = 3) were analyzed and scored according to their phenotype into two categories (puncta/even). Used antibodies and corresponding dilutions are listed in Dataset EV5.

## FRAP

RPE cells were seeded to μ-Slide 8 Well (Ibidi) chambers at a density of 10,000 cells per cm² 24 h prior to transfection. Cells were transfected with corresponding plasmids of ECFP-tagged DVL3 construct (Datasets EV2 and EV3) and incubated for another 24 h. Live-cell imaging and photobleaching was performed on a Zeiss LSM 880 microscope equipped with a thermostat and $CO_2$ chamber using Plan-Apochromat ×63 objective. After transferring of cells into the microscope, they were left for 30 min to rest. Then, protein condensates of similar size were bleached using a 458-nm laser and images were taken every 5 s. Only condensates that stayed in the plane of focus were considered for analysis. Raw images were processed using ImageJ software using jay plugins (https://github.com/jayunruh/Jay_Plugins) from Jay Unruh at Stowers Institute for Medical Research in Kansas City, MO. Plugins used: create spectrum jru v1, combine all trajectories jru v1, normalize trajectories jru v1 (Min_Max normalization), batch FRAP fit jru v1 and average trajectories jru v1; all using default settings.

## LC-MS analysis of polyglutamylated DVL

Selected 1D gel bands were excised manually and after destaining and washing procedures each band was subjected to protein reduction (10 mM DTT in 25 mM NaHCO₃, 45 min, 56 °C, 750 rpm) and alkylation (55 mM IAA in 25 mM NaHCO₃; 30 min, laboratory temperature, 750 rpm). After further washing by 50% ACN/NaHCO₃ and pure ACN, the gel pieces were incubated with 125 ng trypsin (sequencing grade; Promega) in 50 mM NaHCO₃. The digestion was performed for 2 h at 40 °C on a Thermomixer (750 rpm; Eppendorf). Tryptic peptides were extracted into LC-MS vials by 2.5% formic acid (FA) in 50% ACN with the addition of polyethylene glycol (20,000; final concentration 0.001%, Stejskal et al, 2013) and concentrated in a SpeedVac concentrator (Thermo Fisher Scientific). Additional digestion of selected tryptic digests was conducted by adding 5.2 nmol of CNBr to a vacuum-dried digest reconstituted in 0.5 M HCl (16 h, 25 °C, 750 rpm).

LC-MS/MS analyses of peptide mixtures coming from in-gel digestions were done using RSLCnano system (Thermo Fisher Scientific) online connected to Impact II Qq-Time-Of-Flight mass spectrometer (Bruker, Bremen, Germany). Prior to LC separation, tryptic digests were online concentrated and desalted using trapping column (100 µm × 30 mm, 40 °C) filled with 3.5-µm X-Bridge BEH 130 C18 sorbent (Waters, Milford, MA, USA). After washing of trapping column with 0.1% FA, the peptides were eluted (flow 300 nl/min) from the trapping column onto an Acclaim Pepmap100 C18 column (3 µm particles, 75 µm × 500 mm, 40 °C; Thermo Fisher Scientific, Waltham, MA, USA) by 50 min nonlinear gradient program (1–56% of mobile phase B; mobile phase A: 0.1% FA in water; mobile phase B: 0.1% FA in 80% acetonitrile). The analytical column outlet was directly connected to the CaptiveSpray nanoBooster ion source (Bruker). NanoBooster was filled with acetonitrile and nanoBooster pressure was set to 0.2 Bar.

MS data were acquired in a data-dependent strategy with 3 s long cycle time. Mass range was set to 150–2200 *m/z* and precursors were selected from 300–2000 *m/z*. Acquisition speed of MS and MS/MS scans was 2 Hz and 4–16 Hz, respectively. Default CID collision energies and isolation widths with respect to precursor charge and *m/z* were used.

Under equivalent ionization and fragmentation conditions, peptide standards at a concentration 0.01 mg/ml (in 50% ACN, 0.1% FA) were directly injected at 3 µl/min flow rate to the ion source by using a syringe pump.

The processing of the mass spectrometric data including recalibration, compounds detection, charge deconvolution and further MS/MS data analysis was carried out in DataAnalysis software (4.2 SR1; Bruker). Mascot MS/MS ion searches (Matrixscience, London, UK; version 2.5.1) were done against in-house database containing expected protein sequences extended with polyE motifs added to the protein C-term or introduced into the sequence after sites tentatively forming side chains. cRAP contaminant database (downloaded from http://www.thegpm.org/crap/) was searched in advance to exclude contaminant spectra prior the main database search. Mass tolerance for peptides and MS/MS fragments were 10 ppm and 0.1 Da, respectively, with the option of one 13C atom to be present in the parent ion. Oxidation of methionine, carbamidomethylation (C), deamidation (N, Q) were set as optional modifications. In case of CNBr digests, methionine modification to homoserine or homoserine lactone was

also allowed. All searches were done without enzyme specificity. Presence of modified C-terminal peptides was verified by checking the corresponding extracted ion chromatograms ( ± 0.015 *m/z*) and by manual verification of the MS/MS data obtained. The mass spectrometry proteomics data have been deposited to the ProteomeXchange Consortium via the PRIDE (Perez-Riverol et al, 2022) partner repository with the dataset identifier PXD034237.

## LC-MS analysis of protein complexes

LC-MS/MS analyses of all peptide mixtures were done using RSLCnano system connected to Orbitrap Fusion Lumos mass spectrometer (Thermo Fisher Scientific). Prior to LC separation, tryptic digests were online concentrated and desalted using trapping column (100 µm × 30 mm) filled with 3.5-µm X-Bridge BEH 130 C18 sorbent (Waters). After washing of trapping column with 0.1% FA, the peptides were eluted (flow rate—300 nl/min) from the trapping column onto an analytical column (Acclaim Pepmap100 C18, 3 µm particles, 75 µm × 500 mm; Thermo Fisher Scientific) by 100 min nonlinear gradient program (1–56% of mobile phase B; mobile phase A: 0.1% FA in water; mobile phase B: 0.1% FA in 80% ACN). Equilibration of the trapping column and the column was done prior to sample injection to sample loop. The analytical column outlet was directly connected to the Digital PicoView 550 (New Objective) ion source with a sheath gas option and SilicaTip emitter (New Objective; FS360-20-15-N-20-C12) utilization. ABIRD (Active Background Ion Reduction Device, ESI Source Solutions) was installed (Paclíková et al, 2021).

MS data were acquired in a data-dependent strategy with cycle time for 3 s and with survey scan (350–2000 *m/z*). The resolution of the survey scan was 60,000 (200 *m/z*) with a target value of $4 \times 10^5$ ions and maximum injection time of 50 ms. HCD MS/MS (30% relative fragmentation energy, normal mass range) spectra were acquired with a target value of $5.0 \times 10^4$ and resolution of 30,000 (200 *m/z*). The maximum injection time for MS/MS was 50 ms. Dynamic exclusion was enabled for 60 s after one MS/MS spectra acquisition. The isolation window for MS/MS fragmentation was set to 1.6 *m/z* (Tanasa et al, 2023).

The analysis of the mass spectrometric RAW data files was carried out using the MaxQuant software (version 1.6.0.16) using default settings unless otherwise noted. MS/MS ion searches were done against modified cRAP database (based on http://www.thegpm.org/crap) containing protein contaminants like keratin, trypsin etc., and UniProtKB protein database for Homo Sapiens (https://ftp.uniprot.org/pub/databases/uniprot/current_release/knowledgebase/reference_proteomes/Eukaryota/UP000005640/UP000005640_9606.fasta.gz; downloaded 22.11.2017, version 2017/11, number of protein sequences: 21,009). Oxidation of methionine and proline, deamidation (N, Q), and acetylation (protein N-terminus) as optional modification, and trypsin/P enzyme with 2 allowed miscleavages were set. Peptides and proteins with FDR threshold <0.01 and proteins having at least one unique or razor peptide were considered only. Match between runs was set among all analyzed samples. Protein abundance was assessed using protein intensities calculated by MaxQuant (Tanasa et al, 2023). The mass spectrometry proteomics data have been deposited to the ProteomeXchange Consortium via the PRIDE (Perez-Riverol et al, 2022) partner repository with the dataset identifier PXD033548.

## Interactome analysis

Protein intensities reported in proteinGroups.txt file (output of MaxQuant) were further processed using the software container environment (https://github.com/OmicsWorkflows), version 3.7.2a. Processing workflow is available upon request. Briefly, it covered: (a) removal of decoy hits and contaminant protein groups, (b) protein group intensities $\log_2$ transformation, (c) LoessF normalization, (d) imputation by the global minimum and (e) differential expression using LIMMA statistical test. The volcano plot was created in R version 3.6.1, using the log fold change and adjusted $P$ value reported by LIMMA. Proteins passing the threshold of log fold change >1 and adjusted $P$ value <0.05 were further processed by REPRINT with integrated CRAPOME and SAINTexpress tools. For the background list of contaminants, CC66 was used; SAINTexpress was run with the default settings only against the CRAPOME control. REPRINT output was further processed by ProHits-Viz web interface producing Dot Plots and Correlation heatmaps using default settings. Scripts to reproduce analyses are available at Zenodo https://doi.org/10.5281/zenodo.11197196.

## Recombinant protein purification

Full-length DVL3 wt, DVL3-E, DVL3 DCA (E710A/D714A) and TTLL11 (212-800) were produced as described in (31) with minimal changes: Twin-Strep-Flag-HALO N-terminally tagged DVL3 or TTLL11 were expressed in HEK293 cells using transient transfection. Cells were harvested 48 h post-transfection, resuspended in a lysis buffer (50 mM Tris pH 8, 150 mM NaCl, 10 mM KCl, 10% glycerol) with cocktail of protease inhibitors (#11836145001, Roche) and 0.2% NP40 (#74385, Sigma). The mixture was incubated for 20 min on ice and cell lysis was enhanced by dounce. Cell lysate was cleared by centrifugation at $100{,}000 \times g$ for 45 min at 4 °C and supernatant was loaded on Strep-Tactin Superflow high-capacity column (#2-1237-001, IBA) equilibrated in the lysis buffer. The column was washed in lysis buffer, and the protein was eluted using lysis buffer supplemented with 3 mM desthiobiotin. Eluted proteins were concentrated to 0.5 mg/ml using protein concentrators (#88516; Thermo Fisher Scientific), flash frozen in liquid nitrogen, and aliquots were stored at −80 °C. Tubulin and assembled MT were prepared as previously described (Skultetyova et al, 2017).

## Peptide cloning, expression, and purification

pET vectors expressing C-terminal peptides of Dvl3 (C24 or C24-12E) were constructed using annealed oligo cloning. Complementary oligonucleotides were designed to create a double-stranded fragment with 5′ overhangs compatible with NcoI and KpnI restriction enzymes sites. Overlapping oligonucleotides were annealed, phosphorylated at 5′-termini by T4 Polynucleotide Kinase (M0201, New England Biolabs) and ligated using T4 DNA Ligase (M0202, New England Biolabs) to the pET vector digested with the same restriction enzymes. Finally, the expression constructs of the following form were obtained: $His_6$–ZZ tag–TEV cleavage site–peptide sequence. Peptides were expressed in *E. coli* BL21(DE3) strain (C2527I, New England Biolabs) and grown in M9 medium for $^{15}$N labeling. The expression was induced with 0.5 mM IPTG (Alchimica). After cell harvest and lysis, the supernatant was

loaded on an IMAC column (HisTrap HP 5 ml, GE Healthcare) in a buffer containing 25 mM Tris pH 8.0, 500 mM NaCl, 10 mM imidazole, 5% glycerol and eluted in the same buffer with 500 mM imidazole. The eluant was treated with TEV protease (produced in-house) for 1 h at 25 °C to cleave the tag and dialyzed for 16 h at 4 °C to remove the imidazole. The next day, the sample was boiled, snap cooled, and spined at $4000 \times g$ for 20 min at 25 °C before loading to IMAC for tag removal. The flow-through containing the peptide was concentrated by centrifugation (Vivaspin 20, 3000 MWCO PES, Sartorius) and subjected to SEC (Superdex 75 10/300 GL, GE Healthcare) in NMR buffer (50 mM phosphate buffer pH 6.5, 50 mM KCl). Peptides were detected in SEC using low UV wavelength (214 nm). The CK1ε kinase core (1–301) was expressed in LB medium and purified as described before for CK1δ kinase core (Long et al, 2012).

## Nuclear magnetic resonance spectroscopy

All experiments were recorded at 25 °C. Chemical shift assignment of the $^{15}$N-labeled peptides was performed using a combination of $^{15}$N-NOESY-HSQC and $^{15}$N-TOCSY-HSQC experiments recorded at a 950 MHz Bruker Avance NEO spectrometer. Phosphorylation kinetics were measured using a series of $^1$H-$^{15}$N HSQC experiments at a 700 MHz Bruker Avance NEO spectrometer. The general sample composition of a real-time phosphorylation measurement was as follows: $^{15}$N-labeled peptide (50 μM), ATP (1 mM), $MgCl_2$ (10 mM), $d_6$-EDTA (1 mM), $D_2O$ for lock (10%), CK1ε kinase core (1 μM; kinase:peptide ratio of 1:50) in NMR buffer up to a final volume of 200 μl. Kinase was added last in the NMR tube and series of $^1$H-$^{15}$N HSQC spectra were recorded every 30 min over a time course of 16 h.

## Nuclear magnetic resonance kinetic analysis

NMR spectra were analyzed using Sparky 3.115. The fraction of phosphorylation for a given amino acid at a given time point was calculated by dividing the peak intensity of the phosphorylated form by the sum of peak intensities of phosphorylated and non-phosphorylated forms ($I_{phospho}/(I_{phospho} + I_{nonphospho})$). The experimental data were fitted to a mono-exponential function $f(x) = A*(1-\exp(-k*x))$ using Gnuplot 5.2.4.

## In vitro polyglutamylation

DVL3 was mixed with TTLL11 (212-800) in BRB30 Buffer (30 mM PIPES, 8 mM MgCl2, 1 mM EGTA, pH 6.8) supplemented by 4 mM ATP, 4 mM Glu, 2 mM TCEP, 2×PI, 1 mM EDTA, TEV protease 0.03 mg/ml. Final protein concentration was 0.75 μM DVL, 0.75 μM MTs and 0.15 μM TTLL11. The mixture was either mixed with the sampling buffer or incubated 60 min at 37 °C and subsequently mixed with the sampling buffer. The proteins were separated using SDS-PAGE and visualized by CBB staining or immunoblotting. MS analysis was performed on the samples from excised bands of interest from CBB-stained gel.

## Statistics

In graphs, numerical data are shown as mean ± SD as depicted. One-way ANOVA with Dunnett' multiple comparisons test was used for dual-luciferase assay in Fig. EV4C. For comparisons of fold

change, one-sample $t$ test with theoretical mean = 1 was used as samples were normalized to control (= 1) which as a result had zero variability (*$P \leq 0.05$; **$P \leq 0.01$; ***$P \leq 0.001$; ns, not significant, $P > 0.05$). Indicated statistical tests were calculated by GraphPad Prism 9 (GraphPad Software Inc.).

For DVL3 subcellular localization experiments, the data on frequencies puncta vs. even were analyzed using linear mixed model fit by REML and Type III ANOVA with Satterthwaite's method followed by multiple comparisons of means (Tukey contrast) in R software version 3.6.1 using lme4 and multcomp packages. Other method-specific statistical operations are explained in the corresponding method section. Scripts to reproduce analyses are available at Zenodo https://doi.org/10.5281/zenodo.11197197.

## Animal models

The work with *Xenopus laevis* and *Zebrafish* was carried out according to the Czech animal use and care law and approved by the local authorities and committees (Animal Care and Housing Approval: 45055/2020-MZE-18134, issued by Ministry of Agriculture of the Czech Republic; and Animal Experiments Approval: CZ 62760214, issued by State Veterinary Administration/Section for South Moravian Region).

The adult *Xenopus laevis* frogs used for embryo generation are housed in the local husbandry facility at Masaryk University in a XenopLus system (catalog number RE18001301, Tecniplast, Italy), which is an advanced, fully automated system designed for amphibians, ensuring excellent animal care and housing conditions. The Xenopus laevis adults were purchased from the European Xenopus Research Center (ERXC) in 2021.

*Zebrafish* strain AB (ZDB-GENO-960809-7) was kept in the Techniplast ZebTec automatic recirculating system. They are fed twice a day with Artemia salina and Zebrafeed dry food by Sparos. The animals live under a 14/10-h day/night cycle and are kept in 28 °C water. After collecting the eggs and injections, the embryos were kept in E3 medium in an incubator at 28.5 °C until 14.5 hpf.

Only *Xenopus* and *Zebrafish* embryos at early developmental stages were studied, and no adult animals were involved in these experiments.

## Xenopus embryos

The generation and cultivation of Xenopus embryos were performed following general protocols. In brief, testes from males under anesthesia (20% MS-222, Sigma-Aldrich, #A5040) were removed surgically from the body cavity and transferred to cold 1× Marc's Modified Ringers (MMR; 100 mM NaCl, 2 mM KCl, 1 mM MgSO$_4$, 2 mM CaCl$_2$, 5 mM HEPES, buffered to pH 7.4) supplemented with 50 μg/mL of gentamycin (Sigma-Aldrich, #G3632). To induce egg laying, fully mature Xenopus females were injected with 260 U of human chorionic gonadotropin (hCG; Merck, Ovitrelle 250 G) into the dorsal lymph sac about 12–16 h before use and kept overnight at 18 °C. For fertilization, eggs were squeezed from an induced female directly into a petri dish and mixed with a piece of testes in 0.1× MMR. After 20 min, embryos were dejellying using 3% cysteine solution (Sigma-Aldrich, C7880) buffered to pH 7.8 for circa 10 min until they contact each other. Washed embryos (5× in 0.1× MMR solution) were cultivated at 16–18 °C for subsequent microinjections of nucleic acids.

The RNA for microinjections was synthesized from the plasmid pCS2 myc-xDvl3 WT and its derivate 12E using the mMessage mMachine SP6 kit (Thermo Fisher, #AM1340). xTTLL11 ATG MO (taactttcttatgggcttttcccag) and xTTLL11 splicing MO for exon 3 (tacagaataatgagggccactca) were ordered from Gene tools (https://www.gene-tools.com/). For microinjections, embryos were cultivated in 3% Ficoll PM400 (Cytiva, #17030010) in 0.6× MMR and injected at the four-cell stage into the two dorsal blastomeres with 100 pg of myc-xDvl3 RNAs for the analysis of morphogenetic defects. Before gastrulation (stage 10.5), the embryos were cultivated in 0.1× MMR until they reached stage 26, fixed for 2 h in formaldehyde solution 4% buffered to pH 6.9 (Sigma-Aldrich, #100496), and analyzed for morphogenetic defects. Alternatively, five embryos (stage 26) were lysed before fixation in a 5× Laemmli loading buffer for the subsequent SDS-PAGE/WB analysis. As for the analysis of blastopore closure, the embryos were cultivated in 0.1× MMR enriched with 50 μg/mL of gentamycin (Sigma-Aldrich, #G3632). The number of embryos for each condition is indicated below the graphs (created by GraphPad software) and it is representative of 1–3 biological experiments. All embryos were staged after the normal table of Niewkoop and Faber (Nieuwkoop and Faber, 1994). The pictures of embryos were taken on plates with 2% agarose (Serva, #11404) using Leica S9i stereoscope and LAX software.

## Zebrafish crispants

Oligonucleotide sequences for gRNA production were selected using the web tool https://chopchop.cbu.uib.no (Labun et al, 2019). Sequences of gRNA and oligos for sequencing are in the Dataset EV6. A protocol published by Shah et al (Shah et al, 2015) was utilized for gRNA production. A mixture consisting of rCas9NLS protein (0.66 mg/ml, Sigma-Aldrich CAS9PL), Phenol red (0.05%, Sigma-Aldrich, P0290), 120 mM KCl, and gRNA (200 ng/μl) was injected in a volume of 1 nl into the cells of one-cell stage embryos of the AB zebrafish strain. In control injections, an equivalent concentration of BSA was used in place of rCAS9 in an otherwise identical mixture. At 14.5 hpf, dechorionated embryos were fixed using 4% PFA for one hour, followed by three 20-minute washes in PBST (1% Tween in PBS). Subsequently, samples were permeabilized for 10 min in 1% Triton X-100 in PBS, washed for 20 min in PBST and blocked for 2 h in 3% BSA in PBST. Following blocking, the myoD1 antibody (Abcam ab209976, 1:400) was added and incubated overnight at 4 °C on a slow shaker. Then, samples were washed 3× for 20 min in PBST and incubated with a secondary antibody (α-Rabbit Alexa Fluor 488, 1:500) for 2 h at room temperature. After incubation, a final round of washing was performed. Finally, samples were dissected, mounted on slides, and imaged using a Leica SP8 confocal microscope.

## Graphics

Graphics in Figs. 2B, 4A, 5A, 5E, 6F, EV4A, EV4, and synopsis image were created with BioRender.com

# Data availability

The datasets and computer code produced in this study are available in the following databases: LC-MS analysis of DVL3 polyglutamylation data: PRIDE PXD034237; Polyglutamylated

DVL3 interactome data: PRIDE PXD033548; Scripts to reproduce analysis: Zenodo 11197197. All reagents used are mentioned in "Methods" or listed in supplementary material with corresponding source and/or RRID. Unique reagents generated in this study (plasmids, recombinant proteins) are available upon request.

The source data of this paper are collected in the following database record: biostudies:S-SCDT-10_1038-S44318-024-00254-7.

## Peer review information

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

## Acknowledgements

Received funding from Czech Science Foundation grant GX19-28347X (VB); French National Research Agency grant ANR-10-IDEX-0001-02 (CJ); LabEx Cell'n'Scale grant ANR-11-LBX-0038 (CJ); Institut de convergence Q-life grant ANR-17-CONV-0005 (CJ); French National Research Agency grant ANR-12-BSV2-0007 (CJ); French National Research Agency grant ANR-17-CE13-0021 (CJ); Czech Academy of Sciences grant RVO: 86652036 (CB); Czech Science Foundation (Project GA22-25365S) (CB); Czech Science Foundation (Project GA23-07149S) (CB); Czech Science Foundation (Project GA22-06405S) (JH); and Charles University Grant Agency (project no. 1414120) (JN). The authors acknowledge the core facility CELLIM supported by the Czech-BioImaging large RI project (LM2018129 funded by MEYS CR) for their support with obtaining scientific data presented in this paper. CIISB research infrastructure project LM2018127 funded by MEYS CR and European Regional Development Fund-Project "UP CIISB" (No. CZ.02.1.01/0.0/0.0/18_046/0015974 are gratefully acknowledged for the financial support of the measurements at the Proteomics Core Facility (CEITEC) and at the Josef Dadok National NMR Centre. Computational resources were supplied by the project "e-Infrastruktura CZ" (e-INFRA LM2018140) provided within the program Projects of Large Research, Development and Innovations Infrastructures The project National Institute for Cancer Research (Programme EXCELES, ID Project No. LX22NPO5102) - Funded by the European Union – Next Generation EU. The authors would like to thank Bryjalab members and Mikołaj Nowak (Institut Curie) for technical support.

## Author contributions

**Marek Kravec**: Conceptualization; Data curation; Formal analysis; Validation; Investigation; Visualization; Methodology; Writing—original draft;

Writing—review and editing. **Ondrej Šedo**: Data curation; Software; Formal analysis; Investigation; Methodology; Writing—review and editing. **Jana Nedvědová**: Funding acquisition; Investigation; Writing—review and editing. **Miroslav Micka**: Software; Formal analysis; Investigation. **Marie Šulcová**: Formal analysis; Investigation; Visualization. **Nikodém Zezula**: Formal analysis; Investigation; Visualization. **Kristína Gömöryová**: Data curation; Software; Formal analysis; Visualization; Methodology; Writing—review and editing. **David Potěšil**: Formal analysis; Methodology. **Ranjani Sri Ganji**: Formal analysis. **Sara Bologna**: Resources. **Igor Červenka**: Conceptualization; Writing—review and editing. **Zbyněk Zdráhal**: Supervision; Writing—review and editing. **Jakub Harnoš**: Supervision; Methodology. **Konstantinos Tripsianes**: Supervision. **Carsten Janke**: Resources; Supervision; Funding acquisition; Writing—review and editing. **Cyril Bařinka**: Supervision; Funding acquisition; Writing—review and editing. **Vítězslav Bryja**: Conceptualization; Supervision; Funding acquisition; Methodology; Writing—original draft; Project administration; Writing—review and editing.

Source data underlying figure panels in this paper may have individual authorship assigned. Where available, figure panel/source data authorship is listed in the following database record: biostudies:S-SCDT-10_1038-S44318-024-00254-7.

## Disclosure and competing interests statement

The authors declare no competing interests.

# Expanded View Figures

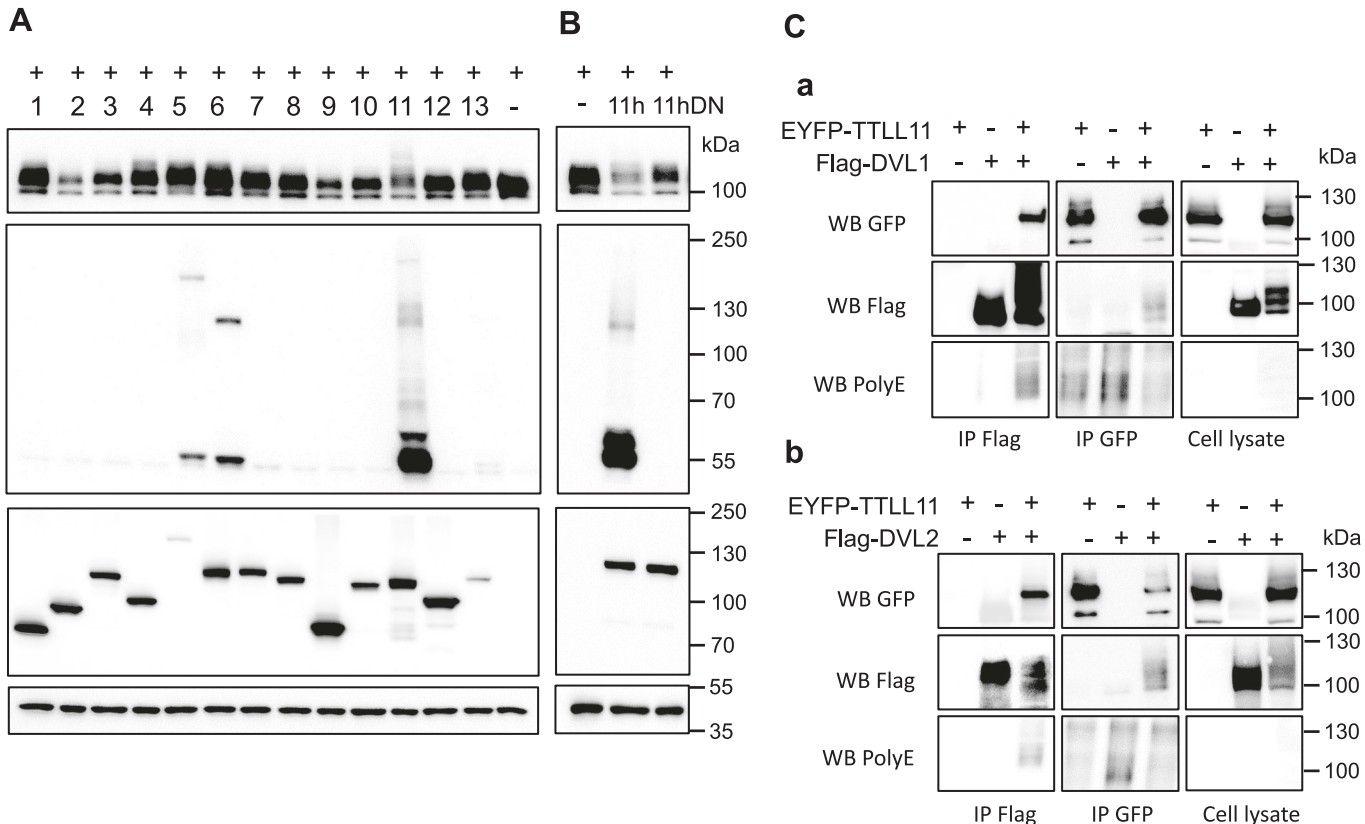

**Figure EV1.   TTLL11 binds and polyglutamylates DVL1 and DVL2.**

(**A, B**) HEK293T cells were transfected with constructs encoding murine YFP tagged TTLL1 – TTLL13 (**A**) or with human TTLL11 and its inactive variant TTLL11 (E531G; DN) (**B**) together with Flag-DVL2. The samples were subjected to WB analysis using polyglutamylation specific antibody—PolyE. Note the appearance of polyE-positive bands of DVL2 size in conditions where TTLL11 and DVL2 were co-expressed. (**C**) Co-immunoprecipitation of Flag-DVL1 (**a**) or Flag-DVL2 (**b**) with TTLL11 overexpressed in HEK293T cells. TTLL11 was co-immunoprecipitated with both DVL1 and DVL2 and both DVLs were polyglutamylated in the pulldown when co-expressed with TTLL11 (IP Flag, WB PolyE). Source data are available online for this figure.

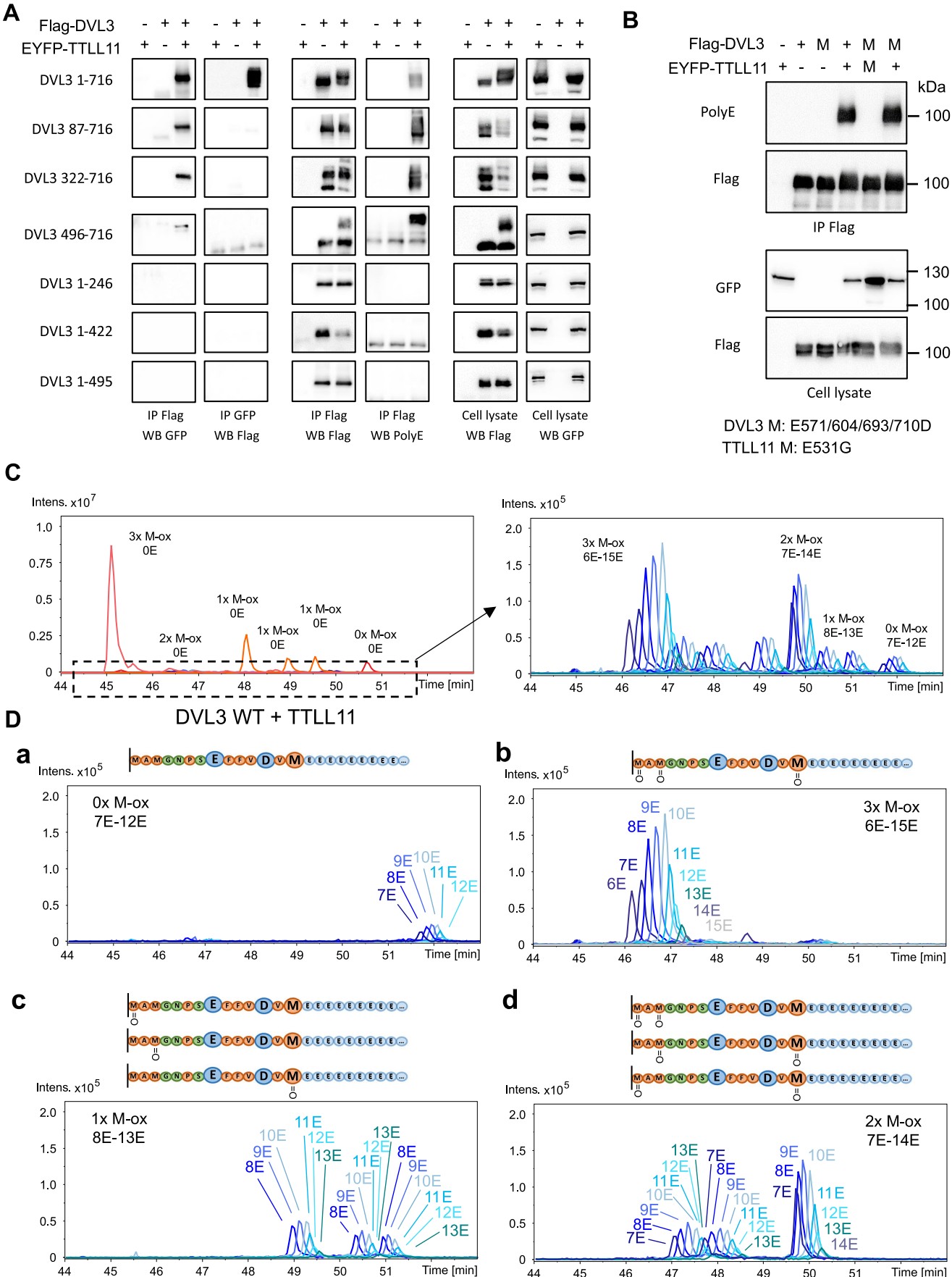

DVL3 M: E571/604/693/710D
TTLL11 M: E531G

DVL3 WT + TTLL11

**Figure EV2. Modification site of DVL3 polyglutamylation.**

(A) Domain mapping of DVL3 polyglutamylation and search for TTLL11 interaction domain on DVL3. DVL3 and its truncated mutants were co-expressed with TTLL11 in HEK293T cells and DVL3 was subsequently immunoprecipitated. Only the DVL3 mutants containing C-terminal part were able to pull down TTLL11. Samples were also stained by modification-specific antibody PolyE, which detected polyglutamylation in all DVL3 mutants containing its C-terminus. (B) Polyglutamylation of DVL3 E571D/E604D/E693D/E710D is comparable to DVL3 wt. DVL3 variants were overexpressed with TTLL11 or with its inactive mutant TTLL11 E531G in HEK293 cells and immunoprecipitated via Flag tag and analyzed by WB as indicated (C, D) EIC chromatogram shows peaks corresponding to the very last C-terminal peptide of DVL3 formed after tryptic cleavage and its polyglutamylated variants, that are highlighted in a separate window. The data are from the same experiment as Fig. EV3A. (C) EIC shows peaks corresponding to polyglutamylated peptides shown schematically above each chromatogram. (D) chromatograms for non- (Da), mono- (Dc), di- (Dd) and tri-oxidized (Db) peptides are presented separately. Source data are available online for this figure.

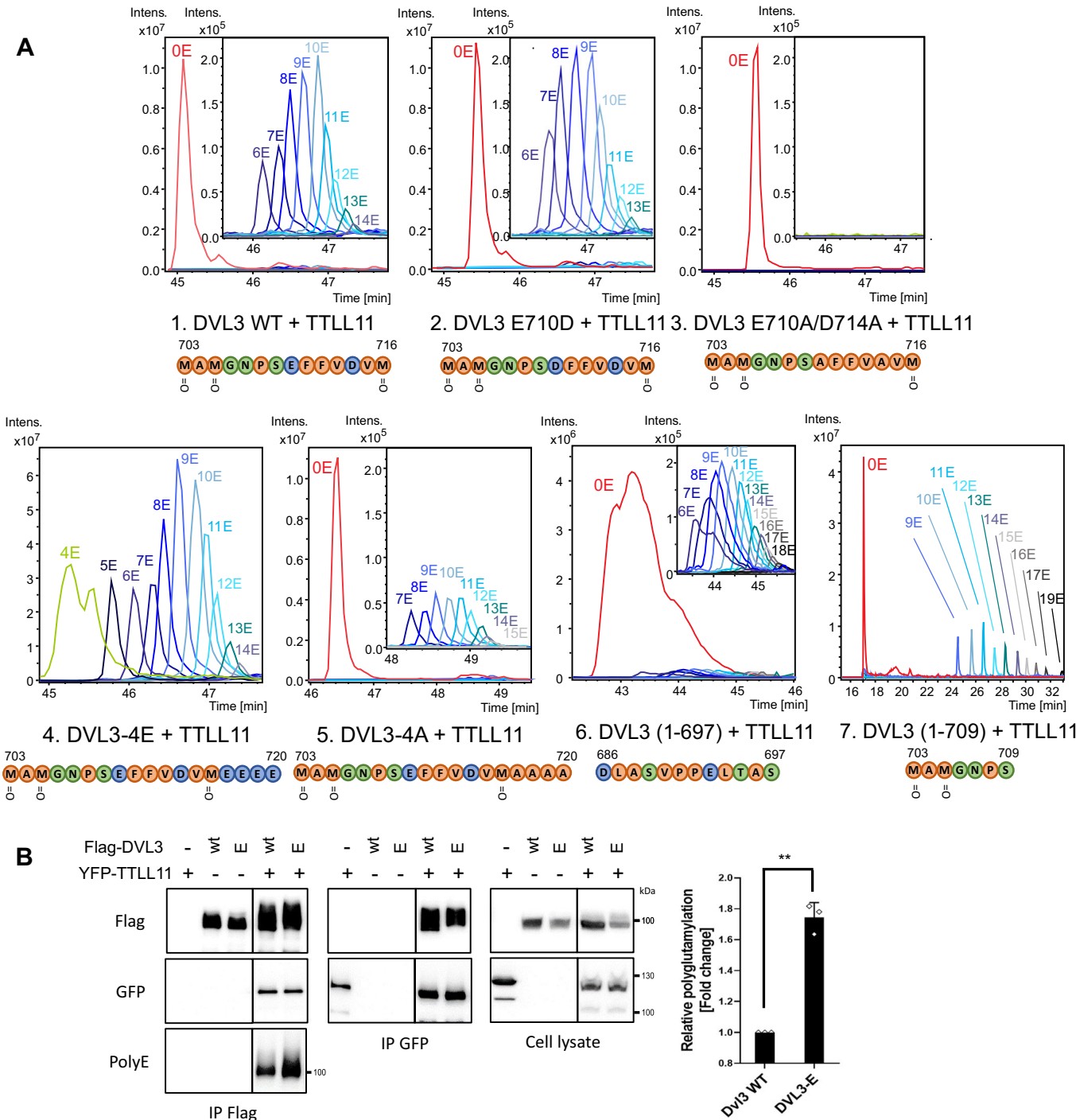

**Figure EV3. C-terminal polyglutamylation.**

(A) EIC showing polyglutamylation of DVL3 C-terminus by TTLL11 in WT protein and mutated variants from Fig. 3A. Lower intensity peaks for polyglutamylated peptides are highlighted in separate windows. Results are shown for fully M-ox peptides. (B) Polyglutamylation of Flag-DVL3 C-terminus after addition of 1E residue at the C-terminal M716. Relative polyglutamylation was derived from PolyE band intensity, normalized to total protein (Flag) signal, and is shown as a fold change compared to Dvl3 WT polyglutamylation in three biological replicates. Graph represents mean with SD. Statistical significance was assessed using one-sample $t$ test with theoretical mean = 1; **$P$ = 0.0055. Source data are available online for this figure.

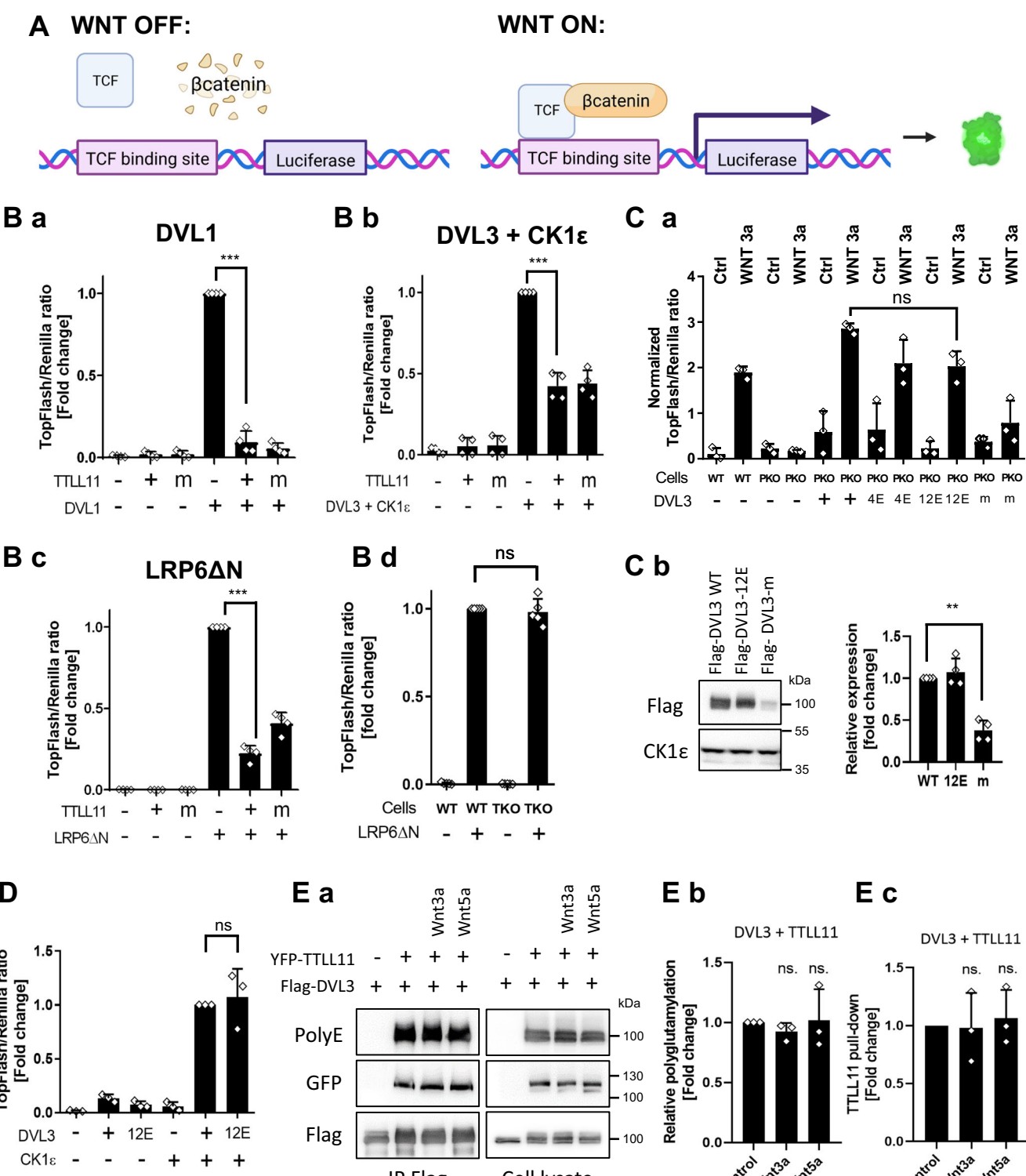

◀ **Figure EV4.   DVL3 polyglutamylation does not affect activity in the Wnt/β-catenin pathway.**

(A) TopFlash method scheme. WNT/β-catenin pathway activation results in induction of TCF target genes, and in production of active luciferase. (B) TopFlash reporter assay of TTLL11 and its inactive mutant E531G (indicated as m) with Wnt/β-catenin pathway activators: DVL1 (Ba), DVL3 + CK1 (Bb) and LRP6ΔN (Bc). (Bd) TopFlash reporter assay of LRP6ΔN in Hek293 T-rex (WT) or HEK293T-rex DVL1/TVL2/DVL3 KO cells (TKO). The data are represented as fold change compared to the inducer sample. *** represents (Ba) $P = 0.0001$; (Bb) $P = 0.0008$; (Bc) $P < 0.001$, ns not significant. (C) DVL3 rescue assays in HEK293T-rex RNF43/ZNRF3/DVL1/DVL2/DVL3 penta knockout cells (PKO) that lack endogenous DVL and cannot respond to Wnt3a. (Ca) PKO cells were transfected by indicated DVL3 variant and Wnt/β-catenin pathway signaling was induced by Wnt3a. WT cells = control. (Cb) DVL3m, DVL3-WT and DVL3-12E expression in cells transfected by the same amount of DNA. WB intensities are normalized to DVL3 WT; $n = 4$. CK1ε = loading control, **$P = 0.002$ (D) Both DVL3 and DVL3-12E potently activate TopFlash assay upon CK1ε co-expression. (Ea) Analysis of DVL3 polyglutamylation and interaction with TTLL11 after stimulation by Wnt3a or Wnt5a. WB intensities for polyglutamylation (PolyE; Eb) or TTLL11 co-purification (GFP; Ec) were normalized to Flag signal (DVL3 amount). Results from biological three replicates are shown as a fold change to control polyglutamylation (Eb) or pulldown (Ec). TopFlash data represent mean ± SD; $n = 4$ for B; $n = 5$ for (C); $n = 3$ for (D), ns = not significant. Statistics: one-sample *t* test with theoretical mean = 1 for (B, Cb, D, Eb, E); one-way ANOVA with Dunnett' multiple comparisons test for Ca. (B, D, E) were performed in HEK293T cells. Source data are available online for this figure.

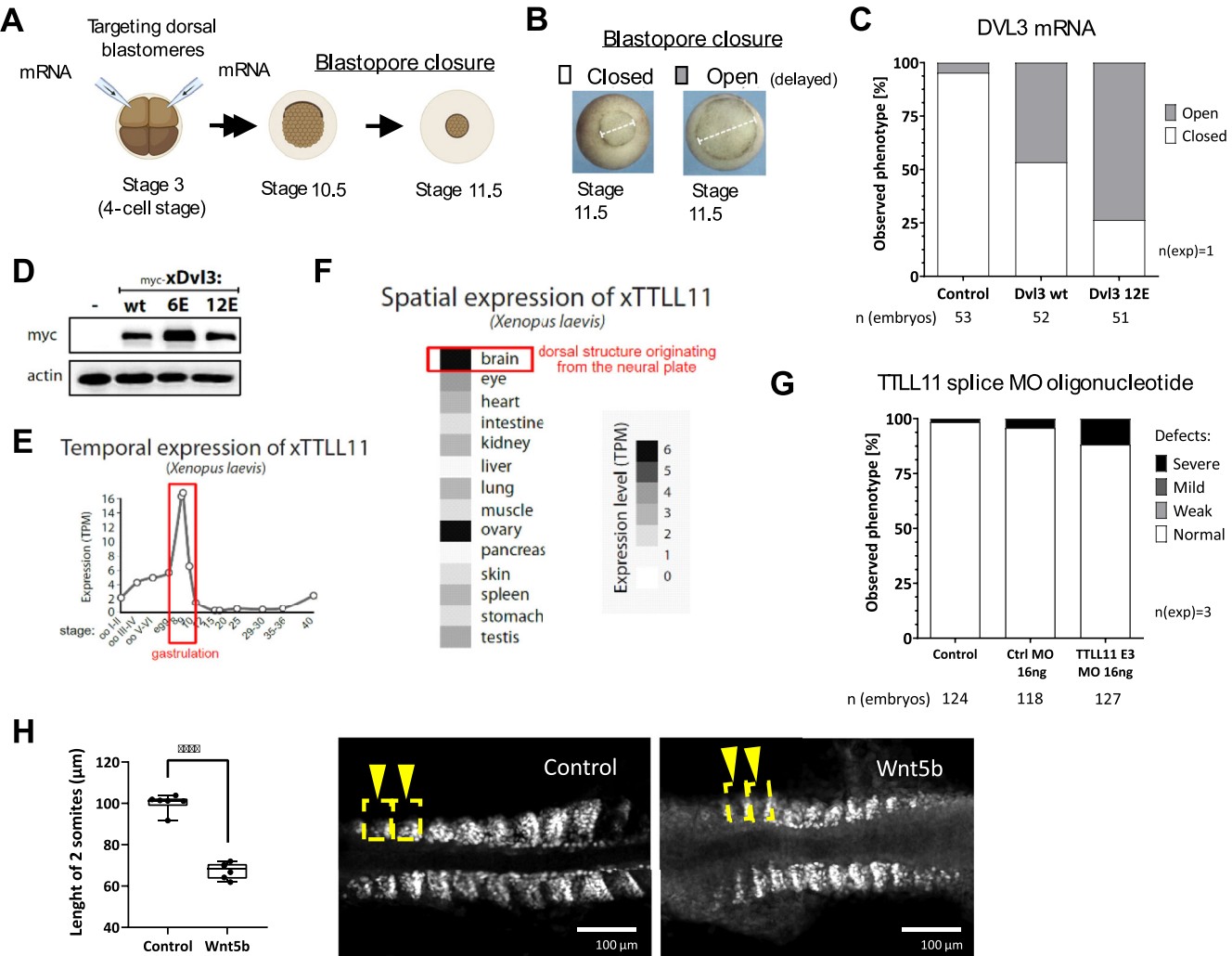

**Figure EV5. Role of DVL3 polyglutamylation in *Xenopus laevis* and *Danio rerio* embryonal development.**

(A) Both dorsal blastomeres of 4-cell *Xenopus laevis* embryo were injected with xDvl3 mRNA and embryos were observed during blastopore closure at stage 11.5. (B) Normal or delayed blastopore closure was assessed at stage 11.5. (C) Effect on uninjected embryos and embryos injected with either mRNA of xDvl3 WT or xDvl3-12E modification mimicking mutant. (D) WB analysis of xDvl3 protein amount in *Xenopus* embryo lysates. (E) xTTLL11 RNA expression during the development of *Xenopus laevis*. (F) Spatial expression of xTTLL11 in adult organism (E, F adapted from (Session et al, 2016)). (G) Effect of splicing MO targeting xTTLL11 exon 3 on gastrulation and neurulation (see also Fig. 5D). (H) The length of the first 2 anterior somites was measured for control embryo Wnt5b KO embryos (*n* = 6). Box plots are shown as median (middle bar) with 25th and 75th percentiles and whiskers showing min to max values. Statistical analysis was performed by unpaired *t* test; ****$P$ < 0.0001. Source data are available online for this figure.

