## [Peer Review File · The EMBO Journal]

Carboxy-terminal polyglutamylation regulates signaling and phase separation of the Dishevelled protein

Marek Kravec, Ondrej Šedo, Jana Nedvědová, Miroslav Micka, Marie Šulcová, Nikodém Zezula, Kristína Gömöryová, David Potesil, Ranjani Sri Ganji, Sara Bologna, Igor Cervenka, Zbynek Zdrahal, Jakub Harnoš, Konstantinos Tripsianes, Carsten Janke, Cyril Barinka, and Vitezslav Bryja

Corresponding author: Vitezslav Bryja (bryja@sci.muni.cz)

Review Timeline:

Submission Date:	13th Nov 23
Editorial Decision:	1st Feb 24
Revision Received:	30th Jun 24
Editorial Decision:	7th Aug 24
Revision Received:	15th Aug 24
Accepted:	16th Sep 24

Editor: Ieva Gailite

Transaction Report:

Dear Vita,

Thank you for submitting your manuscript for consideration by the EMBO Journal. We have now received comments from three reviewers, which are included below for your information.

As you can see, reviewers #2 and #3 are largely positive in their assessment, with reviewer #3 pointing out several reasonable aspects of improvement. However, reviewer #1 is rather critical and indicates that further evidence for DVL3 glutamylation and phase separation at the endogenous level would be needed. Furthermore, he/she finds that the in vivo phenotypes currently cannot be clearly attributed to DVL3 glutamylation.

Based on the positive assessments of reviewers #2 and #3 and your input during the pre-decision consultation, I would like to invite you to revise your manuscript according to the provided revision plan, including adding the analysis of polyglutamylation and TTLL11 colocalisation of DVL3 at endogenous or endogenous-like levels. Please also add additional discussion on DVL3 phase separation and on the complexity of the in vivo phenotypes due to the potential redundancy with other TTLL family proteins. If it is feasible to test Ttll11 knockout phenotypes in zebrafish with another set of guide RNAs, I think it would be helpful to attempt this line of investigation.

I should also add that it is The EMBO Journal policy to allow only a single major round of revision and that it is therefore important to resolve the main concerns at this stage.

We generally allow three months as standard revision time, which can be extended up to six months. As a matter of policy, competing manuscripts published during this period will not negatively impact on our assessment of the conceptual advance presented by your study. However, please contact me as soon as possible upon publication of any related work to discuss the appropriate course of action. I have currently extended the deadline to four months. Should you foresee a problem in meeting this deadline, please let me know in advance to discuss an extension.

When preparing your letter of response to the referees' comments, please bear in mind that this will form part of the Review Process File and will therefore be available online to the community. For more details on our Transparent Editorial Process, please visit our website: <https://www.embopress.org/page/journal/14602075/authorguide#transparentprocess>. Please also see the attached instructions for further guidelines on preparation of the revised manuscript.

Please feel free to contact me if you have any further questions regarding the revision. Thank you for the opportunity to consider your work for publication. I look forward to discussing your revision.

With best regards,

Ieva

At EMBO Press we ask authors to provide source data for the main manuscript figures. Our source data coordinator will contact

you to discuss which figure panels we would need source data for and will also provide you with helpful tips on how to upload and organize the files.

We realize that it is difficult to revise to a specific deadline. In the interest of protecting the conceptual advance provided by the work, we recommend a revision within 4 months (1st April 2024). Please discuss the revision progress ahead of this time with the editor if you require more time to complete the revisions.

Referee #1:

This manuscript reports the glutamylation of the protein Dishevelled by the TTLL11. Previous proteomic studies have shown that many cellular proteins are glutamylated, in addition to tubulin which is the more prominent substrate for this modification. However, very little is known about the function of glutamylation on non-tubulin targets. The authors provide in vitro and cellular overexpression data pertaining to this interaction and examine phenotypes in two model systems, *Xenopus* and Zebrafish. The major problem with this study is that all the cellular assays rely on overexpression of both TTLL11 and DVL3. Under these conditions it is easy to drive the interaction and modification of this protein. Given the negative charges that glutamylation adds to any substrate, it is not surprising that the interactome for overexpressed DVL3 changes in response to glutamylation, but it is very concerning that the DVL interactome has very little overlap with previously published ones (possibly again due to driving non-physiological interaction because of the overexpression). The LLPS data is again also obtained with overexpressed proteins. The experiments in *Xenopus* gastrulation also do not offer support that the modification of DVL3 by TTLL11 is relevant in vivo and responsible for the phenotype observed. The authors show that TTLL11 loss has a clear phenotype both in *Xenopus* and Zebrafish, but again do not have evidence to link this phenotype to the modification of DVL3. Different approaches of editing the modification sites in the endogenous DVL3 protein as well as evidence of interaction between these proteins without their overexpression are needed to support the premise of this study. Therefore, it is my opinion that this work is not suitable for publication.

Referee #2:

This paper by Kravec et al (Bryja lab) is an interesting study on the role of polyglutamylation within the C-terminal region of Dishevelled (Dsh/Dvl), here focusd on DVL3, for its function. The authors show convincingly that post-translational modification of reversible polyglutamylation is required for normal DVL3 function. The tubulin tyrosine ligase-like (TTLL) enzyme family catalyzes the modification, and it is shown here that TTLL11 generates polyglutamylation at a glutamate residue. TTLL11 polyglutamylates DVL3, and thus changes its interactome. Furthermore, this increases its capacity for phosphorylation, which itself triggers a liquid-liquid phase separation (LLPS), required for non-canonical Wnt-signaling. The modification was reversible by a deglutamylating enzyme (CCP6). At the in vivo functional level the authors show that convergence extension gastrulation and neurulation events are affected both by a KD of TTLL11 (via morpholinos) and similarly that a *Xenopus* DVL3 protein mutated in the glutamylation target residues affects the process as well. Overall, it is shown that this posttranslational modification, new to Dsh-family proteins, broadens the range of proteins that are polyglutamylated and it provides evidence that polyglutamylation can act as a regulator of liquid-liquid phase separation of proteins.

Dsh-family proteins are key components of the core planar cell polarity (PCP) interaction network/pathway, often referred to as non-canonical Wnt/PCP signaling. The addition of a novel post-translational modification to the Dsh family of proteins, and PCP "signaling" in general, is of significance and of interest to the field and cell biology of signaling in general. This is further augmented by the demonstration that canonical Wnt/b-catenin signaling is not affected.

The authors use a broad range of technology applications from hard core biochemical and biophysical assays to cell culture and to in vivo studies. The presented data and conclusions are of high quality and, again, general interest in the cell polarity field and in developmental and cell biology in general. I thus fully support the publication of the paper in EMBO Journal. I have reviewed an earlier version of this paper for another high-profile journal and the current version is much improved and so I do not suggest additional experimental data to further improve the presentation. It is an impressive data set!

However, the paper would benefit from a Conclusions section that extracts and summarizes the key findings. As presented the Discussion is a bit chatty and while it mentions a lot of interesting points, a clean Conclusion statement including a mention of the limitations of the study is missing. For example, it should be noted that a lack of rescue experiments in the in vivo settings prevents a clear in vivo function analysis and/or the paper lacks a substantive analysis of potential PCP defects in tissues like the skin or the inner ear, where PCP has been best defined in vertebrates.

Other comment:

The referencing to Wnt/PCP signaling is marginal with one reference in the introduction (ref 14: review by Butler and Wallingford 2017). As all functional data focus on PCP signaling this needs to be expanded, and credit should be given to all the ground-breaking work in *Drosophila* that established the role of Dsh/Dvl proteins in Wnt/PCP signaling.

I suggest the authors add three original references for Dsh/Dvl role in PCP signaling:

- Klingensmith J, Nusse R, Perrimon N. (1994) *Genes Dev.* 8(1):118-30. PMID: 8288125
- Boutros M, Paricio N, Strutt DJ, Mlodzik M. (1998) *Cell.* 94(1):109-18. PMID: 9674432
- Wallingford JB, Rowling BA, Vogeli KM, Rothbacher U, Fraser SE, Harland RM. (2000) *Nature.* 405(6782):81-5. PMID: 10811222

And they should also add a couple recent reviews on Wnt/PCP signaling from the *Drosophila* and mammalian fields:

- Devenport D. (2014) *J Cell Biol.* 207(2):171-9. PMID: 25349257
- Humphries AC, Mlodzik M. (2018) *Curr Opin Cell Biol.* 51:110-116. PMID: 29289896
- Harrison C, Shao H, Strutt H, Strutt D. (2020) *Biochem Soc Trans.* 48(4):1297-1308. PMID: 32820799
- Koca Y, Collu GM, Mlodzik M. (2022) *Curr Top Dev Biol.* 150:255-297. PMID: 35817505

Referee #3:

General Summary and Significance

In this manuscript, Kravec et al. describe experiments showing that the backbone carboxyl group of the C-terminal methionine of DVL3 can be polyglutamylated by TTL11. Glutamylated DVL3 occurs with a positive feedback mechanism allowing polyglutamylation to occur. Polyglutamylation changes the DVL3 interactome, among other changes, enhancing an interaction with CK1epsilon that leads to enhanced phosphorylation of DVL3. Of note, both polyglutamylation and phosphorylation appear to decrease the number of cells with DVL3-containing puncta suggesting that both processes inhibit formation of biomolecular condensates by DVL3. Polyglutamylation also enhances the fluidity of these biomolecular condensates. The deglutamylating enzyme, CCP6, was able to promote a punctate cellular pattern for DVL3, reinforcing the role of glutamylation in modulating condensate formation by DVL3. In this wide-ranging paper, the authors also demonstrate that polyglutamylation controls DVL3 activity in the planar cell polarity pathway, but not in Wnt/beta-catenin signalling.

The data presented in Fig. S6, showing the failure to demonstrate a specific effect of glutamylation on Wnt/beta-catenin signalling, are quite valuable, and reflect the overall rigor and quality of the work, with multiple hypotheses presented, experiments performed to test them, and data shown disproving or supporting the hypotheses. This approach was very much appreciated. The study includes data from multiple models (cell, animal model, in vitro) and using multiple approaches, biochemistry, mass spectrometry, NMR and more. Overall, this is a well executed study and the conclusions are supported by the presented data.

Importantly, mass spectrometry data provide evidence that the glutamylation occurs on the carboxy group of the C-terminal methionine, and results point to a significant impact of modification on condensate formation and properties. This is intriguing and suggests that glutamylation may be more common than now appreciated, with a significant role in regulating biomolecular condensates, pointing to broad implications of the work.

There are no major concerns essential to support the conclusions.

Concerns that should be addressed

The DVL3 "interactome" is defined via co-IP pull-down. This is in contrast to the recognition that interactomes of proteins found within biomolecular condensates, as DVL isoforms are described to be, are often missing in co-IP analysis due to the transient interactions that underlie condensate formation. Proximity labeling approaches are increasingly performed as they are expected to more completely define interactomes within such condensates. It has also been suggested that the CRAPOME database of co-IP contaminants is highly correlated with propensity to phase separate. Thus, the REPRINT analysis using the CRAPOME dataset to remove "contaminants" may, in fact, be removing actual interactors. While proximity labeling studies are likely beyond the scope of the work, they should be discussed in the context of limitations to this study in defining the interactome, and a secondary REPRINT analysis without removing CRAPOME "contaminants" could also be performed.

The authors should propose a mechanism by which polyglutamylation and phosphorylation inhibit phase separation. One assumes an electrostatic imbalance with significantly more negative charges than positive charges would play a role, but this should be explored, even if only in the discussion.

The description of "liquid-liquid phase separation" is too specific for the likely more complex phase transitions that occur with biological molecules, particularly in cells (DOI: 10.1016/j.molcel.2022.05.018, DOI: 10.1073/pnas.2210177119 and other papers). This terminology should be changed to simply "phase separation" to reflect that complexity.

Papers by Schwartz-Romond et al. (DOI: 10.1038/nsmb1247, DOI: 10.1242/jcs.02646) suggest a correlation between the ability of DVL protein to form puncta and signalling in the Wnt pathway. The authors should rationalize why polyglutamylation-based disruption of DVL puncta does not affect signalling in the Wnt pathway.

Additional non-essential suggestion

Endogenous DVL proteins are known to form puncta under certain conditions. The authors demonstrate that when overexpressed TTLL11, which partitions into DVL3 puncta, is spread through the cytoplasm and on filamentous structures, but does not appear in a punctate pattern. Do endogenous DVL proteins form puncta in HEK293T cells or can they be induced to do so? Would overexpressed TTLL11 partition into endogenous DVL puncta under these conditions? Or is it necessary to have highly overexpressed DVL proteins to see TTLL11 partitioning? Can the overexpressed TTLL11 dissolve puncta formed by endogenous DVL3 puncta due to the high stoichiometric ratio?

Response to the editor and reviewers:

Editor:

Thank you for submitting your manuscript for consideration by the EMBO Journal. We have now received comments from three reviewers, which are included below for your information. As you can see, reviewers #2 and #3 are largely positive in their assessment, with reviewer #3 pointing out several reasonable aspects of improvement. However, reviewer #1 is rather critical and indicates that further evidence for DVL3 glutamylation and phase separation at the endogenous level would be needed. Furthermore, he/she finds that the in vivo phenotypes currently cannot be clearly attributed to DVL3 glutamylation. Based on the positive assessments of reviewers #2 and #3 and your input during the pre-decision consultation, I would like to invite you to revise your manuscript according to the provided revision plan, including adding the analysis of polyglutamylation and TTLL11 colocalisation of DVL3 at endogenous or endogenous-like levels. Please also add additional discussion on DVL3 phase separation and on the complexity of the in vivo phenotypes due to the potential redundancy with other TTLL family proteins. If it is feasible to test Ttll11 knockout phenotypes in zebrafish with another set of guide RNAs, I think it would be helpful to attempt this line of investigation.

Response: In the revised version of the manuscript we have addressed all the points requested by the editor. These include analysis of co-localization of TTLL11/DVL3/polyglutamylation at close to endogenous levels (novel Fig. 8), improved the discussion (aspects of DVL3 phase separation and its physiological counterparts, possible MT compartments where TTLL11/DVL3 meet, redundancy with other TTLL family proteins). We have also added another set of Ttll11 crispants that has confirmed the original conclusion. In addition, we have also addressed other points of the reviewers as specified below in the detailed response to reviewers.

Referee #1:

This manuscript reports the glutamylation of the protein Disheveled by the TTLL11. Previous proteomic studies have shown that many cellular proteins are glutamylated, in addition to tubulin which is the more prominent substrate for this modification. However, very little is known about the function of glutamylation on non-tubulin targets. The authors provide in vitro and cellular overexpression data pertaining to this interaction and examine phenotypes in two model systems, Xenopus and Zebrafish. The major problem with this study is that all the cellular assays rely on overexpression of both TTLL11 and DVL3. Under these conditions it is easy to drive the interaction and modification of this protein. Given the negative charges that glutamylation adds to any substrate, it is not surprising that the interactome for overexpressed DVL3 changes in response to glutamylation, but it is very concerning that the DVL interactome has very little overlap with previously published ones (possibly again due to driving non-physiological interaction because of the overexpression). The LLPS data is again also obtained with overexpressed proteins. The experiments in Xenopus gastrulation also do not offer support that the modification of DVL3 by TTLL11 is relevant in vivo and responsible for the phenotype observed. The authors show that TTLL11 loss has a clear phenotype both in Xenopus and Zebrafish, but again do not have evidence to link this phenotype to the modification of DVL3. Different approaches of editing the modification sites in the endogenous DVL3 protein as well as evidence of interaction between these proteins without their overexpression are needed to support the premise of this study. Therefore, it is my opinion that this work is not suitable for publication.

Response: As any other study, our work also has, unfortunately, some limitations. We acknowledge these in the revised version of the study, but do not see many ways how to overcome them.

The key points of referee #1 refer in general to the issue of (I) possible technical artifacts caused by overexpression both with respect to the interaction, polyglutamylation, and LLPS, and (II) physiological relevance. Below we copy points of the referee #1 and address these two issues separately.

I) Artifacts of overexpression.

“Reviewer: The major problem with this study is that all the cellular assays rely on overexpression of both TLL11 and DVL3. Under these conditions it is easy to drive the interaction and modification of this protein. The LLPS data is again also obtained with overexpressed proteins.”

It is true that the mechanistic part of the study depends at some point on overexpression of both DVL3/TLL11. However, we would like to point out that we could clearly prove that overexpressed TLL11 can also polyglutamylate endogenous DVL2 and DVL3 (see Fig. 1E). Unfortunately, fully endogenous setup for co-IP is not technically possible because of the combination of very low abundance of proteins in question and the quality of available antibodies. We have intensely explored this line of experimenting over many years with the conclusion that it is not feasible in our hands.

In general, we disagree with the referee that *“it is easy to drive the interaction and modification of this protein”*. There is multiple evidence of this fact presented in the study:

- The polyglutamylation was identified uniquely at the C-terminal carboxylate of DVL3 (no promiscuity towards carboxylate groups of internal glutamates as in the case of tubulin)
- Under identical experimental conditions we were able to identify a DVL3 mutant that cannot be modified, despite it interacts and co-localizes with TLL11.
- Similarly, only one CCP family member could efficiently remove polyglutamylation, despite the fact that other CCP members also co-localized with DVL3 in puncta.
- We have also reconstituted the polyglutamylation *in vitro* with purified components and can show that it is a highly regulated reaction.

With respect to the specificity of LLPS, we consider very important our finding that the extent of polyglutamylation in cells, which could be modulated by distinct complementary approaches (TLL11 WT vs. MUT; DVL3 WT x polyE x non-glutamylated; minus/plus CCP6 deglutamylase), was in all these combinations, independently of the molecular nature of the trigger, reflected in LLPS features of DVL3. The combined evidence clearly suggests that indeed the effects are specific.

“Reviewer: Given the negative charges that glutamylation adds to any substrate, it is not surprising that the interactome for overexpressed DVL3 changes in response to glutamylation, but it is very concerning that the DVL interactome has very little overlap with previously published ones (possibly again due to driving non-physiological interaction because of the overexpression).”

This comment of the reviewer is unfortunately based on the misunderstanding. In the DVL3 interactome section, we show a change in DVL3 interactome after co-expression with TLL11 (DVL3 vs DVL3/TLL11). In DVL3 co-IPs we have indeed identified many known interactors of DVL3, known from literature (Axin1, CK1 isoforms, CK2, Vangl1, DVL1 and DVL2, etc). These data accompany the manuscript and are deposited on PRIDE (accession no PXD033548) for further inspection.

II) Physiological relevance of TLL11-mediated DVL3 polyglutamylation.

“ Reviewer: The experiments in Xenopus gastrulation also do not offer support that the modification of DVL3 by TLL11 is relevant in vivo and responsible for the phenotype observed. The authors show that TLL11 loss has a clear phenotype both in Xenopus and Zebrafish, but again do not have evidence to link this phenotype to the modification of DVL3. Different approaches of editing the modification sites in the endogenous DVL3 protein as well as evidence of interaction between these proteins without their overexpression are needed to support the premise of this study.”

- The in vivo part in its nature depends on the combination of multiple approaches and multiple angles (each with its own limitations) that synergistically support each other. We provide three independent line of evidence that support our conclusion that indeed polyglutamylation of DVL3 is relevant for non-canonical Wnt signaling in vivo: (i) DVL3 variants gain of function in Xenopus, (ii) TLL11 loss of function in Xenopus and (iii) loss of function of putative effectors identified by mass spec (Katnal2, RAB11FIP5) in zebrafish.
- We would like to highlight that despite the suggested editing of the DVL3 C-terminus in vivo can provide additional insights, it could NOT deliver a clear-cut experiment: It is well known (by work us and others) that mutations at the DVL3 C-terminus will affect not only polyglutamylation but also open/close conformation of DVL3 with all critical consequences for DVL3 function. Of note, the C-termini of DVL3 are not fully conserved among human, *Xenopus* and zebrafish. In order to be able to interpret the possible phenotypes in non-human experimental models, we would need to repeat the mechanistic part performed with human DVL3/TLL11 and characterize DVL3 variants suitable for loss-of-function experiments.

After the discussion with the editor during the pre-decision consultation, we have addressed the criticism of reviewer #1 as follows:

- We have dedicated a paragraph in the discussion to clearly acknowledge the limitations of our study (page 14, lines 624 - 644)
- We have added data (novel Fig. 8) where we describe (co-)localization of polyglutamylation, TLL11 and close to endogenous DVL3 (page 11, lines 471 - 487). This information has been complemented by a thorough summary of the literature of DVL subcellular localization to conclude where polyglutamylation of DVL3 can happen and what can be its biological significance (page 13, line 579 – page 14, line 622).
- We have added a paragraph to discuss the importance of phase separation of DVL, with focus on the evidence obtained using endogenous DVL proteins (page 12, line 544 - page 13, line 577)

In summary, despite we realize the limitations of our study, we are convinced that the main message of our manuscript is neither affected nor invalidated by them. These key novel findings are: (i) novel mechanism of C-terminal polyglutamylation by TLL11, including optimizing of a sensitive, quantitative and qualitative approach for its detection, (ii) proof that polyglutamylation can regulate phase separation of protein condensates, and (iii) TLL11 and polyglutamylation regulate PCP.

Referee #2:

Referee: This paper by Kravec et al (Bryja lab) is an interesting study on the role of polyglutamylation within the C-terminal region of Dishevelled (Dsh/Dvl), here focused on DVL3, for its function. The authors show convincingly that post-translational modification of reversible polyglutamylation is required for normal DVL3 function. The tubulin tyrosine ligase-like (TLL) enzyme family catalyzes the modification,

and it is shown here that TLL11 generates polyglutamylation at a glutamate residue. TLL11 polyglutamylates DVL3, and thus changes its interactome. Furthermore, this increases its capacity for phosphorylation, which itself triggers a liquid-liquid phase separation (LLPS), required for non-canonical Wnt-signaling. The modification was reversible by a deglutamylating enzyme (CCP6). At the in vivo functional level the authors show that convergence extension gastrulation and neurulation events are affected both by a KD of TLL11 (via morpholinos) and similarly that a Xenopus DVL3 protein mutated in the glutamylation target residues affects the process as well. Overall, it is shown that this posttranslational modification, new to Dsh-family proteins, broadens the range of proteins that are polyglutamylated and it provides evidence that polyglutamylation can act as a regulator of liquid-liquid phase separation of proteins.

Dsh-family proteins are key components of the core planar cell polarity (PCP) interaction network/pathway, often referred to as non-canonical Wnt/PCP signaling. The addition of a novel post-translational modification to the Dsh family of proteins, and PCP "signaling" in general, is of significance and of interest to the field and cell biology of signaling in general. This is further augmented by the demonstration that canonical Wnt/b-catenin signaling is not affected.

The authors use a broad range of technology applications from hard core biochemical and biophysical assays to cell culture and to in vivo studies. The presented data and conclusions are of high quality and, again, general interest in the cell polarity field and in developmental and cell biology in general. I thus fully support the publication of the paper in EMBO Journal. I have reviewed an earlier version of this paper for another high-profile journal and the current version is much improved and so I do not suggest additional experimental data to further improve the presentation. It is an impressive data set!

Response: Thank you for the positive evaluation of our work.

Referee: However, the paper would benefit from a Conclusions section that extracts and summarizes the key findings. As presented the Discussion is a bit chatty and while it mentions a lot of interesting points, a clean Conclusion statement including a mention of the limitations of the study is missing. For example, it should be noted that a lack of rescue experiments in the in vivo settings prevents a clear in vivo function analysis and/or the paper lacks a substantive analysis of potential PCP defects in tissues like the skin or the inner ear, where PCP has been best defined in vertebrates.

Response: We would like to thank the reviewer for the suggestion how to improve quality of our discussion. During the revision we have modified our discussion extensively. Among others we have added the final summarizing paragraph (Conclusion) (page 14, lines 624 - 644) that also specifically addresses the limitations of our study.

Referee: The referencing to Wnt/PCP signaling is marginal with one reference in the introduction (ref 14: review by Butler and Wallingford 2017). As all functional data focus on PCP signaling this needs to be expanded, and credit should be given to all the ground-breaking work in Drosophila that established the role of Dsh/Dvl proteins in Wnt/PCP signaling.

I suggest the authors add three original references for Dsh/Dvl role in PCP signaling:

- Klingensmith J, Nusse R, Perrimon N. (1994) Genes Dev. 8(1):118-30. PMID: 8288125*
- Boutros M, Paricio N, Strutt DJ, Mlodzik M. (1998) Cell. 94(1):109-18. PMID: 9674432*
- Wallingford JB, Rowning BA, Vogeli KM, Rothbacher U, Fraser SE, Harland RM. (2000) Nature. 405(6782):81-5. PMID: 10811222*

And they should also add a couple recent reviews on Wnt/PCP signaling from the Drosophila and mammalian fields:

- Devenport D. (2014) *J Cell Biol.* 207(2):171-9. PMID: 25349257
- Humphries AC, Mlodzik M. (2018) *Curr Opin Cell Biol.* 51:110-116. PMID: 29289896
- Harrison C, Shao H, Strutt H, Strutt D. (2020) *Biochem Soc Trans.* 48(4):1297-1308. PMID: 32820799
- Koca Y, Collu GM, Mlodzik M. (2022) *Curr Top Dev Biol.* 150:255-297. PMID: 35817505

Response: We have elaborated the introduction section and added more background information on Wnt/PCP signaling with corresponding references (page 2, lines 69 - 79).

Referee #3:

Referee: In this manuscript, Kravec et al. describe experiments showing that the backbone carboxyl group of the C-terminal methionine of DVL3 can be polyglutamylated by TTL11. Glutamylation occurs with a positive feedback mechanism allowing polyglutamylation to occur. Polyglutamylation changes the DVL3 interactome, among other changes, enhancing an interaction with CK1epsilon that leads to enhanced phosphorylation of DVL3. Of note, both polyglutamylation and phosphorylation appear to decrease the number of cells with DVL3-containing puncta suggesting that both processes inhibit formation of biomolecular condensates by DVL3. Polyglutamylation also enhances the fluidity of these biomolecular condensates. The deglutamylating enzyme, CCP6, was able to promote a punctate cellular pattern for DVL3, reinforcing the role of glutamylation in modulating condensate formation by DVL3. In this wide-ranging paper, the authors also demonstrate that polyglutamylation controls DVL3 activity in the planar cell polarity pathway, but not in Wnt/beta-catenin signalling.

The data presented in Fig. S6, showing the failure to demonstrate a specific effect of glutamylation on Wnt/beta-catenin signalling, are quite valuable, and reflect the overall rigor and quality of the work, with multiple hypotheses presented, experiments performed to test them, and data shown disproving or supporting the hypotheses. This approach was very much appreciated. The study includes data from multiple models (cell, animal model, in vitro) and using multiple approaches, biochemistry, mass spectrometry, NMR and more. Overall, this is a well executed study and the conclusions are supported by the presented data. Importantly, mass spectrometry data provide evidence that the glutamylation occurs on the carboxy group of the C-terminal methionine, and results point to a significant impact of modification on condensate formation and properties. This is intriguing and suggests that glutamylation may be more common than now appreciated, with a significant role in regulating biomolecular condensates, pointing to broad implications of the work. There are no major concerns essential to support the conclusions.

Response: Thank you for the positive evaluation of our work.

Referee: Concerns that should be addressed. The DVL3 "interactome" is defined via co-IP pull-down. This is in contrast to the recognition that interactomes of proteins found within biomolecular condensates, as DVL isoforms are described to be, are often missing in co-IP analysis due to the transient interactions that underlie condensate formation. Proximity labeling approaches are increasingly performed as they are expected to more completely define interactomes within such condensates. It has also been suggested that the CRAPOME database of co-IP contaminants is highly correlated with propensity to phase separate. Thus, the REPRINT analysis using the CRAPOME dataset to remove "contaminants" may, in fact, be removing actual interactors. While proximity labeling studies are likely beyond the scope of the work, they should be discussed in the context of limitations to this study in defining the interactome, and a secondary REPRINT analysis without removing CRAPOME "contaminants" could also be performed.

Response: We would like to thank the reviewer for the suggestions. We agree that proximity labelling could be better approach to catch more transient interactions and also agree that this goes beyond the scope of our work. We are, however, planning a follow up study where we plan exactly the approaches proposed by the reviewer. To allow the alternative analysis (not using CRAPome filters) by others in the field we have deposited the raw data in PRIDE database (accession no. PXD033548) so anyone can access it for further analysis.

Referee: The authors should propose a mechanism by which polyglutamylation and phosphorylation inhibit phase separation. One assumes an electrostatic imbalance with significantly more negative charges than positive charges would play a role, but this should be explored, even if only in the discussion.

Response: We have extended the discussion section (page 12, lines 526 - 537) about mechanism of polyglutamylation /phosphorylation mediated disruption of DVL3 phase separation.

Referee: The description of "liquid-liquid phase separation" is too specific for the likely more complex phase transitions that occur with biological molecules, particularly in cells (DOI: 10.1016/j.molcel.2022.05.018, DOI: 10.1073/pnas.2210177119 and other papers). This terminology should be changed to simply "phase separation" to reflect that complexity.

Response: We have unified our terminology as suggested.

Referee: Papers by Schwartz-Romond et al. (DOI: 10.1038/nsmb1247, DOI: 10.1242/jcs.02646) suggest a correlation between the ability of DVL protein to form puncta and signalling in the Wnt pathway. The authors should rationalize why polyglutamylation-based disruption of DVL puncta does not affect signalling in the Wnt pathway.

Response: Thank you for this suggestion. We have happily expanded this part of the discussion – see page 12, line 544 – page 13 line 577.

Referee: Additional non-essential suggestion. Endogenous DVL proteins are known to form puncta under certain conditions. The authors demonstrate that when overexpressed TLL11, which partitions into DVL3 puncta, is spread through the cytoplasm and on filamentous structures, but does not appear in a punctate pattern. Do endogenous DVL proteins form puncta in HEK293T cells or can they be induced to do so? Would overexpressed TLL11 partition into endogenous DVL puncta under these conditions? Or is it necessary to have highly overexpressed DVL proteins to see TLL11 partitioning? Can the overexpressed TLL11 dissolve puncta formed by endogenous DVL3 puncta due to the high stoichiometric ratio?

Response: Thank you again for this interesting set of questions. In the revised manuscript, we have addressed these questions both experimentally and by improvement in the discussion. We have performed a set of experiments using ECFP-DVL3 at controlled close-to-endogenous levels (see novel Fig. 8 and accompanying text). We have also dedicated a novel paragraph in the discussion that focuses on the situations where DVL can phase separates (ie. forms puncta at the endogenous level). (page 12, line 544 - page 13, line 577)

Dear Vita,

Thank you for submitting a revised version of your manuscript. I sincerely apologise for the protracted assessment process due to delays in referee report submission. We have now received input from one of the original reviewers, who finds that their previous concerns have been addressed satisfactorily. Therefore, there now remain a few editorial points that need addressing before I can extend official acceptance of the manuscript:

1. Please check that the funding information is correct and identical both in the manuscript and our online system. Currently, CELLIM supported by the Czech-Biolmaging large RI project (LM2018129 funded by MEYS CR), CIISB research infrastructure project LM2018127 funded by MEYS CR and European Regional Development Fund-Project „UP CIISB” (No. CZ.02.1.01/0.0/0.0/18_046/0015974, Josef Dadok National NMR Centre, "e-Infrastruktura CZ" (e-INFRA LM2018140), the program Projects of Large Research, Development and Innovations Infrastructures are missing in our system.
2. Please submit up to five keywords.
3. Please make sure that the order of the sections in the manuscript is as follows: abstract, introduction, results, discussion, materials & methods, data availability section, acknowledgments, disclosure statement and competing interests, references, main figure legends, tables, expanded figure legends.
4. There is a reference to partially published data in line 124 (datasets partially published in (de Groot et al, 2014), see Suppl. Table 1). Since our policy does not allow references to data not shown. Please clarify.
5. CRedit has replaced the traditional author contributions section because it offers a systematic, machine-readable author contributions format that allows for more effective research assessment. Please remove the Authors Contributions from the manuscript and use the free text boxes beneath each contributing author's name in our online submission system to add specific details on the author's contribution. More information is available in our guide to authors.
6. Please rename "Competing interests" section into "Disclosure and competing interests statement" (further info: <https://www.embopress.org/page/journal/14602075/authorguide#conflictsofinterest>).
7. Please remove the "Experimental animals" section after "Competing interests" section, as it appears redundant with the information provided in "Materials and Methods".
8. In the legend for figure EV5G, there is a mention for "Suppl. To Fig. 5D". To clarify that this is not a reference to a supplementary figure, please reword, e.g., "see also Fig. 5D".
9. Please rename "Data and materials availability" section into "Data availability" and move it to the end of "Materials and methods" section. Please add resolvable links for PXD034237 and PXD033548 datasets. More information about the format of this section can be found here: <https://www.embopress.org/page/journal/14602075/authorguide#dataavailability>.
10. Appendix Tables should be renamed into Datasets EV1-EV6, and the source files, titles of the legends, callouts in the manuscript should be updated accordingly. Please remove the additional blank sheet in each Excel file. The tables that are not too wide could be moved to the Appendix File - in that case they should be named Appendix Table S1, etc and added to the Appendix PDF file.
11. In the Appendix, please add page numbers and a brief table of contents. Please update the nomenclature to Appendix Figure S1, etc.
12. In our standard image check we noticed that the DVL3 panel in Fig. 6Da does not contain any signal. I realise that it is a negative control, however, some sort of background signal would be expected. Please check.
13. The source data for the following figure panels does not appear to fit to the main figure: Fig 1E, IP DVL3/WB DVL3 blot; Fig 7B, IP FLAG/WB PolyE and IP GFP/WB FLAG. Please check and correct as necessary.
14. In our standard source data check, we have noted unexplained duplicate values figures 5C, 5F and 6C. I have attached the corresponding files with the detected duplications labelled in colour. Please check and correct as needed. A brief explanation would be very helpful.
15. Our data editors have flagged the following issues in figure legends that need correcting:
 - Please define the annotated p values ****/**/* as well as provide the exact p-values for the same in the legend of figure 3b; 5f; 6b; EV 5h; as appropriate.
 - Please provide the exact p values in the legends of figures 6e; 7b-c; EV 3b; EV 4Ba-Bc, Cb.
 - Please indicate the statistical test used for data analysis in the legends of figures 4b; 6b, e; 7c.
 - Please note that the box plots need to be defined in terms of minima, maxima, centre, bounds of box and whiskers, and percentile in the legends of figures 5f; EV 5h.
 - Please note provide information on the nature and number of replicates in the legends of figures 4b; 6e.
 - Please describe the nature of replicates in the legends of figure 3b.
 - Please define the error bars in the legends of figures 3b; 6Cb; 6e; EV 3b.
 - Please define the measure of center for the error bars in the legend of figure 6b.
 - Please define the white arrowheads in the legend of figure 8b-c.

With best wishes,

Ieva

We realize that it is difficult to revise to a specific deadline. In the interest of protecting the conceptual advance provided by the work, we recommend a revision within 3 months (5th Nov 2024). Please discuss the revision progress ahead of this time with the editor if you require more time to complete the revisions.

Referee #3:

In the second version of this manuscript, the authors have largely addressed our concerns with changes to the text.

The study is interesting and well executed and the conclusions are supported by the data. The overall response to reviewer comments is thorough and appropriate, with the revised version of the manuscript more clear. The study is rich with data and strongly supports the proposed novel mechanism of regulation of Dishevelled signalling via polyglutamylation and its impact on phase separation, a very important finding, with potential generality regarding impact of polyglutamylation on regulation.

No major concerns.

Minor concerns:

Figure 7C change "dots" to "punctate" for consistency and clarity.

Clarify or expand on the following sentence in the discussion: "Given the biochemical nature of polyglutamylation that allows for gradual charge and signal modulation - a regulatory principle so far only demonstrated for microtubule severing"

The manuscript requires editing to correct minor grammatical errors.

No other suggestions.

Dear Vita,

Thank you for submitting a revised version of your manuscript. I sincerely apologise for the protracted assessment process due to delays in referee report submission. We have now received input from one of the original reviewers, who finds that their previous concerns have been addressed satisfactorily. Therefore, there now remain a few editorial points that need addressing before I can extend official acceptance of the manuscript:

1. Please check that the funding information is correct and identical both in the manuscript and our online system. Currently, CELLIM supported by the Czech-Biolmaging large RI project (LM2018129 funded by MEYS CR), CIISB research infrastructure project LM2018127 funded by MEYS CR and European Regional Development Fund-Project „UP CIISB" (No. CZ.02.1.01/0.0/0.0/18_046/0015974, Josef Dadok National NMR Centre, "e-Infrastruktura CZ" (e-INFRA LM2018140), the program Projects of Large Research, Development and Innovations Infrastructures are missing in our system.

We have double checked the funding information both in the manuscript and on-line system and added the missing grants to the on-line forms.

2. Please submit up to five keywords.

Dishevelled-3, polyglutamylaton, TTLL11, non-canonical Wnt signalling, protein condensates

3. Please make sure that the order of the sections in the manuscript is as follows: abstract, introduction, results, discussion, materials & methods, data availability section, acknowledgments, disclosure statement and competing interests, references, main figure legends, tables, expanded figure legends.

The manuscript was ordered accordingly.

4. There is a reference to partially published data in line 124 (datasets partially published in (de Groot et al, 2014), see Suppl. Table 1). Since our policy does not allow references to data not shown. Please clarify.

In line with EMBO Journal policy, we have removed the reference.

5. CRediT has replaced the traditional author contributions section because it offers a systematic, machine-readable author contributions format that allows for more effective research assessment. Please remove the Authors Contributions from the manuscript and use the free text boxes beneath each contributing author's name in our online submission system to add specific details on the author's contribution. More information is available in our guide to authors.

Authors contributions were removed from the manuscript.

6. Please rename "Competing interests" section into "Disclosure and competing interests statement" (further info:

<https://www.embopress.org/page/journal/14602075/authorguide#conflictsofinterest>).

Section was renamed.

7. Please remove the "Experimental animals" section after "Competing interests" section, as it appears redundant with the information provided in "Materials and Methods".

Section was removed.

8. In the legend for figure EV5G, there is a mention for "Suppl. To Fig. 5D". To clarify that this is not a reference to a supplementary figure, please reword, e.g., "see also Fig. 5D".

Section was corrected.

9. Please rename "Data and materials availability" section into "Data availability" and move it to the end of "Materials and methods" section. Please add resolvable links for PXD034237 and PXD033548 datasets. More information about the format of this section can be found here:

<https://www.embopress.org/page/journal/14602075/authorguide#dataavailability>.

Section was corrected.

10. Appendix Tables should be renamed into Datasets EV1-EV6, and the source files, titles of the legends, callouts in the manuscript should be updated accordingly. Please remove the additional blank sheet in each Excel file. The tables that are not too wide could be moved to the Appendix File - in that case they should be named Appendix Table S1, etc and added to the Appendix PDF file.

Section was corrected.

11. In the Appendix, please add page numbers and a brief table of contents. Please update the nomenclature to Appendix Figure S1, etc.

Section was corrected.

12. In our standard image check we noticed that the DVL3 panel in Fig. 6Da does not contain any signal. I realise that it is a negative control, however, some sort of background signal would be expected. Please check.

In this case, there was indeed no signal acquired for DVL3 channel, probably by technical mistake. Since it was negative control, the absence of signal did not catch our attention. We have replaced the image for different one from the same experiment with background signal acquired also for DVL3 channel.

13. The source data for the following figure panels does not appear to fit to the main figure: Fig 1E, IP DVL3/WB DVL3 blot; Fig 7B, IP FLAG/WB PolyE and IP GFP/WB FLAG. Please check and correct as necessary.

In Fig 1E, IP DVL3/WB DVL3 blot, there was different exposition used in the source data, it was replaced for the same exposition as is shown in the figure.

In Fig 7B, there are correct images, but as we had available only 15 well gels, and we needed to load 15 samples together to the same gel, the marker was loaded to the first well together with the sample (after previous verification that this approach does not affect the signal). We have added a note to the source data file explaining this approach.

14. In our standard source data check, we have noted unexplained duplicate values figures 5C, 5F and 6C. I have attached the corresponding files with the detected duplications labelled in colour. Please check and correct as needed. A brief explanation would be very helpful.

Figure 5C contained a mistake in calculation. We have corrected the values and added also total number of embryos per each phenotype in addition to the percentage.

Figure 5F contained one duplicated value by copy-paste mistake. It was removed and statistics were recalculated.

Figure 6C contained 2x the same set of values, which were saved to the same file by mistake. Redundant data were not used in the figure and they were deleted from the source data file.

15. Our data editors have flagged the following issues in figure legends that need correcting:

- Please define the annotated p values ****/**/* as well as provide the exact p-values for the same in the legend of figure 3b; 5f; 6b; EV 5h; as appropriate.
- Please provide the exact p values in the legends of figures 6e; 7b-c; EV 3b; EV 4Ba-Bc, Cb.
- Please indicate the statistical test used for data analysis in the legends of figures 4b; 6b, e; 7c.
- Please note that the box plots need to be defined in terms of minima, maxima, centre, bounds of box and whiskers, and percentile in the legends of figures 5f; EV 5h.
- Please note provide information on the nature and number of replicates in the legends of figures 4b; 6e.
- Please describe the nature of replicates in the legends of figure 3b.
- Please define the error bars in the legends of figures 3b; 6Cb; 6e; EV 3b.
- Please define the measure of center for the error bars in the legend of figure 6b.
- Please define the white arrowheads in the legend of figure 8b-c.

Sections of manuscript were corrected. If it was possible, the exact p values were added to the figure legend, otherwise all statistical data were added to the source data files.

Referee #3:

In the second version of this manuscript, the authors have largely addressed our concerns with changes to the text.

The study is interesting and well executed and the conclusions are supported by the data. The overall response to reviewer comments is thorough and appropriate, with the revised version of the manuscript more clear. The study is rich with data and strongly supports the proposed novel mechanism of regulation of Dishevelled signalling via polyglutamylation and its impact on phase separation, a very important finding, with potential generality regarding impact of polyglutamylation on regulation.

No major concerns.

Thank you for positive evaluation of our work.

Minor concerns:

Figure 7C change "dots" to "punctate" for consistency and clarity.

Thank you for the suggestion, we have unified our labelling.

Clarify or expand on the following sentence in the discussion: "Given the biochemical nature of polyglutamylation that allows for gradual charge and signal modulation - a regulatory principle so far only demonstrated for microtubule severing"

We have changed the sentence to be more clear.

The manuscript requires editing to correct minor grammatical errors.

No other suggestions.

Dear Vita,

Thank you for addressing the final editorial issues. I am now pleased to inform you that your manuscript has been accepted for publication in the EMBO Journal. Congratulations on a nice study!

You may qualify for financial assistance for your publication charges - either via a Springer Nature fully open access agreement or an EMBO initiative, e.g., EMBO Press APC support programme for authors from Czech Republic. Check your eligibility: <https://www.embopress.org/page/journal/14602075/authorguide#chargesguide>

If you have any questions, please do not hesitate to contact the Editorial Office. Thank you for your contribution to The EMBO Journal and congratulations on a successful publication!

With best wishes,

leva

leva Gailite, PhD
Senior Scientific Editor
The EMBO Journal
Meyerhofstrasse 1
D-69117 Heidelberg
Tel: +4962218891309
i.gailite@embojournal.org
